# Initial assessment of the spatial learning, reversal, and sequencing task capabilities of knock-in rats with humanizing mutations in the Aβ-coding region of *App*

**Hoa Pham**◉[°], **Tao Yin**◉[°], **Luciano D'Adamio**◉*

Department of Pharmacology, Physiology & Neuroscience New Jersey Medical School, Brain Health Institute, Jacqueline Krieger Klein Center in Alzheimer's Disease and Neurodegeneration Research, Rutgers, The State University of New Jersey, Newark, NJ, United States of America

◉ These authors contributed equally to this work.
* luciano.dadamio@rutgers.edu

**Data Availability Statement:** All relevant data are within the manuscript.

**Funding:** LD Project Number1R01AG073182-01 Contact PI D'ADAMIO, LUCIANO Project

## Abstract

Model organisms mimicking the pathogenesis of human diseases are useful for identifying pathogenic mechanisms and testing therapeutic efficacy of compounds targeting them. Models of Alzheimer's disease (AD) and related dementias (ADRD) aim to reproduce the brain pathology associated with these neurodegenerative disorders. Transgenic models, which involve random insertion of disease-causing genes under the control of artificial promoters, are efficient means of doing so. There are confounding factors associated with transgenic approaches, however, including target gene overexpression, dysregulation of endogenous gene expression at transgenes' integration sites, and limitations in mimicking loss-of-function mechanisms. Furthermore, the choice of species is important, and there are anatomical, physiological, and cognitive reasons for favoring the rat over the mouse, which has been the standard for models of neurodegeneration and dementia. We report an initial assessment of the spatial learning, reversal, and sequencing task capabilities of knock-in (KI) Long-Evans rats with humanizing mutations in the Aβ-coding region of *App*, which encodes amyloid precursor protein (*App*[h/h] rats), using the IntelliCage, an automated operant social home cage system, at 6–8 weeks of age, then again at 4–5 months of age. These rats were previously generated as control organisms for studies on neurodegeneration involving other knock-in rat models from our lab. *App*[h/h] rats of either sex can acquire place learning and reversal tasks. They can also acquire a diagonal sequencing task by 6–8 weeks of age, but not a more advanced serial reversal task involving alternating diagonals, even by 4–5 months of age. Thus, longitudinal behavioral analysis with the IntelliCage system can be useful to determine, in follow-up studies, whether KI rat models of Familial AD (FAD), sporadic late onset AD (LOAD), and of ADRD develop aging-dependent learning and memory deficits.

Number5R01AG063407-02 Contact PI D'ADAMIO, LUCIANO Project Number1RF1AG064821-01 Contact PI D'ADAMIO, LUCIANO All 3 grants are from: National Institutes of Health, National Institute on Aging. USA The funders had no role in study design, data collection and analysis, decision to publish, or preparation of the manuscript.

**Competing interests:** The authors have declared that no competing interests exist.

## Introduction

The model organisms used to model human diseases have major implications on the phenotypic expression of disease-associated genetic mutations. In the past, our laboratory has modeled AD, the prevalent form of dementia among the elderly, and ADRD using a KI approach in mice [1–12]. The KI approach allows to model human diseases in a genetically faithful manner. More recently, we have generated rat KI models of FAD, LOAD, and ADRD [13–19]. The rat is better suited for behavioral tests and other procedures that are important in neurodegenerative diseases' studies. Moreover, gene-expression patterns indicate that rats are better suited to model neurodegenerative diseases. Alternative splicing of *MAPT* [20–23], which is mutated in Frontotemporal Dementia and whose gene product tau forms neurofibrillary tangles (NFT) [24–31], leads to expression of tau isoforms with three or four microtubule binding domains (3R and 4R, respectively). Adult human and rat brains express both 3R and 4R tau isoforms [32]: in contrast, adult mouse brains express only 4R tau [33]. Thus, rats may be a better model organism for dementias with tauopathy.

Aggregated forms of Aβ, a product of APP processing, are, by many, considered the central pathogenic factor in AD. Rat and human APP differ by 3 amino-acids in the Aβ region: given that human Aβ species have higher propensity to form toxic Aβ species as compared to rodent Aβ, we produced rats carrying the humanized Aβ sequence in the endogenous *App* rat alleles ($App^{h/h}$ rats) [13, 14]. This $App^h$ allele allows to study pathogenic mechanisms in FAD, ADRD, and LOAD model organisms producing physiological levels of human Aβ [13–19]. In this view, $App^{h/h}$ rats constitute control animals against which learning and memory performances of our FAD, ADRD, and LOAD models is measured. Whether expression of human Aβ is, per se', sufficient to impact behavior will be addressed in future studies comparing $App^{h/h}$ to $App^{w/w}$ rats.

Behavioral tests are used to determine whether model organisms of AD and ADRD develop learning and memory deficits. Most studies use traditional paradigms, including novel object recognition, fear conditioning, Morris water maze, and radial arm water maze. These approaches are well established and informative. The IntelliCage system (NewBehavior AG) [34, 35] provides an additional method of assessing behavior in rodents. It has been used to study behavior in mouse models of human disease, including neurodegenerative and neuropsychiatric conditions such as Huntington's disease and, notably, AD, with spatial learning and memory being among the most studied parameters [36, 37]. It consists of a central square home cage connected to four operant learning chambers, or corners. Every corner has two sides, each with a drinking bottle gated by a rotating door with a nosepoke sensor (Fig 1). The sides also include LEDs and air puff delivery valves as additional conditioning components. Behavioral programs are defined by the user within a visual coding platform. Subcutaneously injected transponders allow the IntelliCage to track the activity of individual animals with unique radio frequency identification tags. Among the parameters tabulated for subsequent analysis are corner visits, visit lengths, visit times, number of nosepokes per visit, and number of bottle licks per visit. This system offers a variety of advantages over standard cognitive tasks: high-throughput, unbiased data collection; minimal risk of human error; minimal perturbation of testing conditions; and uniform testing of multiple animals simultaneously in a social setting. The final point is important in the context of cognitive phenotyping because the social housing component, a distinguishing feature of this system, eliminates isolation as a confounding psychological stressor from the animals' environment. Its stable, passive manner of data collection also mitigates stress from handling and the traumatic experience inherent in tests such as the Morris water maze [36, 38].

A larger version of the IntelliCage system developed for rats has been used for studying Huntington's disease [39]; the effect of GABA$_B$ receptors in the insula on recognition memory

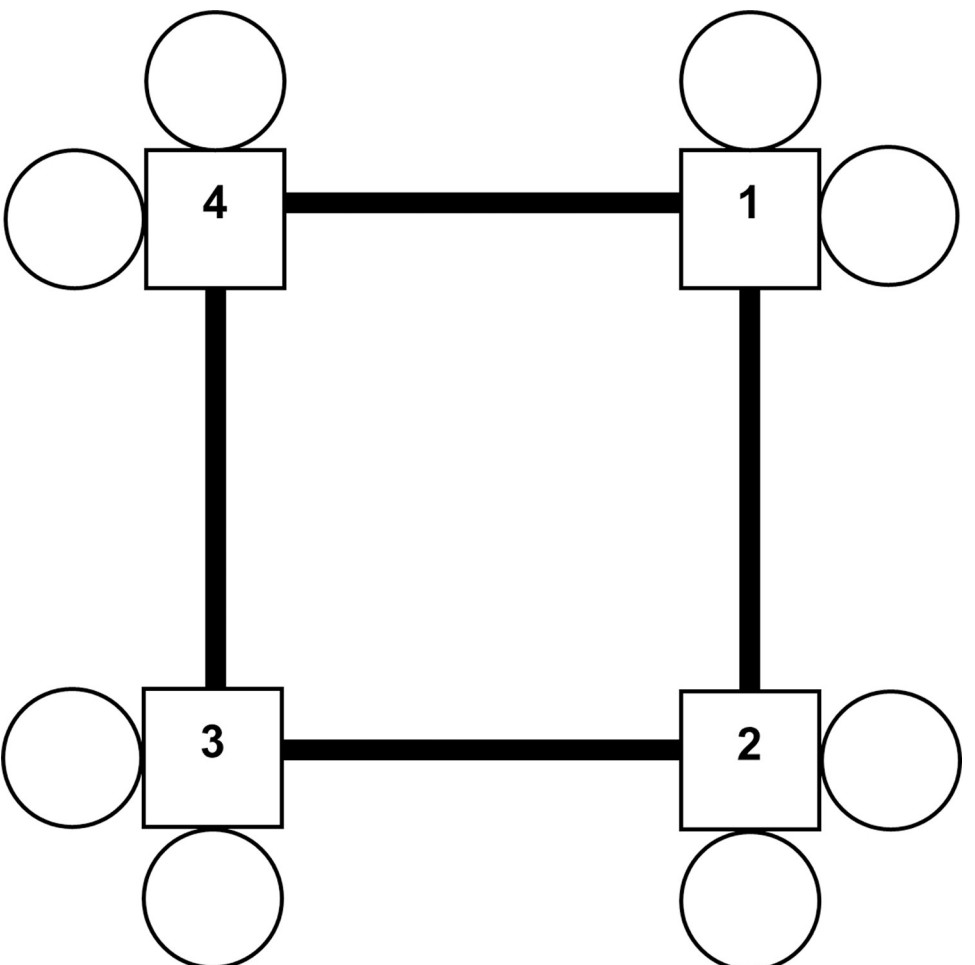

**Fig 1. IntelliCage schematic.** Central home cage and four labeled corners with two drinking sides per corner.

[35]; and deficits in spatial learning and memory following post-weaning social isolation [40]. In this study we tested whether the IntelliCage can be used to assess learning and memory in $App^{h/h}$ rats. We assessed the spatial learning, reversal, and sequencing task performance of male and females $App^{h/h}$ rats at 6–8 weeks of age (peri-adolescent rats), and again at 4–5 months of age (young adult rats). We decided to start testing peri-adolescent rats based on the evidence that our FAD [13], ADRD [19], and LOAD [17, 41] rat models show synaptic plasticity and transmission alterations already at 6–8 weeks of age. Thus, knowing the performance of our control group at this early age is important. We performed a second test on the same animals at 4–5 months of age, to understand how the control $App^{h/h}$ rats perform in longitudinal tests as young adults. We tested both male and female rats to determine whether there are any sex-dependent differences in performance. This is important because incidence rates of LOAD are greater in women than men after age 85 [42]. In summary, this study's goal is to establish methods using the IntelliCage system to determine, in following studies, whether our FAD, ADRD, and LOAD models develop learning and memory deficits, the age of onset of these deficits, whether these deficits correlate with synaptic plasticity/transmission alterations, and the impact of sex.

## Material and methods

### Experimental animals

All experiments were done according to policies on the care and use of laboratory animals of the Ethical Guidelines for Treatment of Laboratory Animals of the NIH. Relevant protocols were approved by the Rutgers Institutional Animal Care and Use Committee (IACUC) (Protocol #201702513). All efforts were made to minimize animal suffering and reduce the number of rats used.

### Rat genotyping and DNA extraction and sequencing

The insertion of humanizing mutations in App exon 16 was verified by DNA sequencing of genomic DNA PCR products that include exon 16 [13]. Fresh cut tail tissue was digested in 300 μl tail lysis buffer (100mM Tris, 5mM EDTA, 0.2% SDS, 200mM NaCl, pH 8.0) with 3 μl of 20 μg/μl protease K at $55^0$ C overnight. 100 μl of protein precipitation solution (7.5M Ammonium Acetate) was added to each sample to precipitate protein. After vortexing samples for 30 seconds, samples were centrifuged at 15000 xg for 5 min. Supernatant containing genomic DNA was mixed with 300 μl Isopropanol. After centrifugation at 15000 xg for 5 min., genomic DNA pellet was desalted with 70% ETOH and was dissolved in water for PCR and sequencing. The rats studied here were obtained by crossing $App^{h/h}$ male and females. Eight breeding pairs were used. To avoid litter effects, for each cohort no more than 2 females and 2 male rats from each breeding pair were used.

### Rat brain preparation

Rats were anesthetized with isoflurane and perfused via intracardiac catheterization with ice-cold PBS. Brains were extracted and homogenized using a glass-teflon homogenizer (w/v = 100 mg tissue/1 mL buffer) in 250 mM Sucrose, 20 mM Tris-base pH 7.4, 1 mM EDTA, 1mM EGTA plus protease and phosphatase inhibitors (ThermoScientific), with all steps carried out on ice or at 4˚C. Samples were flash-frozen in liquid nitrogen immediately after homogenization. Total lysate was solubilized with 1% NP-40 for 30 min rotating at 4˚C. Solubilized lysate was spun at 20,000 g for 10 min, the supernatant was collected and analyzed by ELISA and Western blotting.

### ELISA

Aβ levels were measured with Meso Scale Discovery kit V-PLEX Plus Aβ Peptide Panel 1 (K15200G) according to the manufacturer's recommendations. Human Aβ40 and Aβ42 were revealed using 6E10, a mouse monoclonal antibody raised against human $Aβ_{1-16}$ fragment (BioLegend 803001); rat Aβ40 and Aβ42 were revealed using M3.2, a mouse monoclonal antibody raised against rat/mouse $Aβ_{1-16}$ fragment (Biolegend 11465). Plates were read on a MESO QuickPlex SQ 120. Data were analyzed using Prism software and represented as mean ± SEM.

### Western blot analysis

Protein content quantified by Bradford analysis. Ten μg of protein was brought to 15μl with PBS and LDS Sample buffer-10% β-mercaptoethanol (Invitrogen NP0007) to 1X, boiled for 1 min, cooled on ice, and loaded on a 4–12% Bis-Tris polyacrylamide gel (Biorad 3450125). Proteins were transferred onto nitrocellulose at 25V for 7min using the Trans-blot Turbo system (Biorad) and visualized by red Ponceau staining. Membranes were blocked 45min in 5%-milk (Biorad 1706404), washed extensively in PBS/0.05% Tween20. Primary antibodies were

applied overnight at 4˚C, at 1:1000 dilution in blocking solution (Thermo 37573). Membranes were probed with 6E10 and M3.2. Anti-mouse (Southern Biotech, OB103105) was diluted 1:1000 in 5%-milk and used against mouse and rabbit primary antibodies for 60 min at RT, with shaking. Blots were developed with West Dura ECL reagent (Thermo, PI34076) and visualized on a ChemiDoc MP Imaging System (Biorad).

## Immunohistochemistry (IHC)

Intracardiac PFA-perfused rat brains were extracted and stored in 70% ethanol prior to cerebral coronal sectioning. Sections were dehydrated and paraffin embedded and the processed into 15 cross sections targeting the frontal cortex at the level of the isthmus of the corpus callosum, anterior and posterior hippocampus. IHC staining was performed in accordance with Biospective Standard Operating Procedure (SOP) # BSP-L-06. Slides were manually de-paraffinized and rehydrated prior to the automated IHC. Slides initially underwent antigen retrieval, either heat-induced epitope-retrieval or formic acid treatment. All IHC studies were performed at room temperature on a Lab Vision Autostainer using the REVEAL Polyvalent HRP-AEC detection system (Spring Bioscience). Briefly, slides were incubated sequentially with hydrogen peroxide for 5 minutes, to quench endogenous peroxidase, followed by 5 minutes in Protein Block, and then incubated with primary, antibodies as outlined in Table 1. Antibody binding was amplified using the Complement reagent (20 min), followed by an HRP-conjugate (20 min), and visualized using the AEC chromogen (20 min). All IHC sections were counterstained with Acid Blue 129 and mounted with aqueous mounting medium [43]. The IHC and histology slides were digitized using an Axio Scan.Z1 digital whole-slide scanner (Carl Zeiss). The images underwent quality control (QC) review and final images transferred to the Biospective server for qualitative image analysis.

## Behavioral experiments and analysis

Prior to and after behavioral analysis, males were housed 2 per cage and females were housed 3 per cage under controlled laboratory conditions with a 12-hr dark/light cycle (dark from 7pm to 7am), at a temperature of $25 \pm 1$˚C. They were anesthetized with isoflurane, tagged subcutaneously with radio frequency identification transponders, and allowed to recover for at least a week. Rats had free access to standard rodent diet and tap water while in traditional housing and were monitored for dehydration during periods of water restriction during behavioral analysis. The IntelliCage for Rats (NewBehavior AG) was used to collect behavioral data. Briefly, the program timeline was divided into three parts: (1) a period during which the animals may freely explore the IntelliCage and acclimate to a daily period of restricted water

**Table 1. Primary and amplification antibodies used for IHC.**

| Target | Antibody | Antigen Retrieval | Dilution | Secondary & Amplification |
|---|---|---|---|---|
| Neurons | NeuN, Mouse monoclonal A60, Millipore | Citrate HIER | 1:3000 | RbαM & GtαRb- HRP |
| Amyloid-Beta | 1–16 Aβ, Mouse monoclonal 6E10, Biolegend | 80% Formic Acid | 1:1000 | RbαM & GtαRb-HRP |
| | | | 1:1000 | |
| Microglia | Iba1, Rabbit polyclonal,Wako | Citrate HIER | 1:200 | GtαRb-HRP |
| Astrocytes | GFAP, Rabbit polyclonal, Thermo Scientific | Citrate HIER | 1:200 | DkαRb-bio & SA-HRP |
| Phospho-Tau | pTau, Mouse monoclonal AT8, ThermoScientific | Citrate HIER | 1:1000 | Hα-bio & SA-HRP |

α = anti; bio = biotin; Dk = Donkey; Gt = Goat; HIER = Heat induced antigen retrieval; H = Horse; HRP = Horseradish Peroxidase; M = Mouse; PK = Proteinase K; Rb = Rabbit; SA = Streptavidin.

access during a time window (8:00–11:00pm) called the *drinking session*; (2) a period consisting of place learning and reversal programs during which every animal is assigned a drinking corner during a drinking session; and (3) a period consisting of more complex sequencing programs involving a rule that governs the designation of drinking corners based on animal activity during a drinking session. Variations in the approach toward these parts prompted the design of two parallel cohorts testing the same cognitive domains, analyzed independently. Two cohorts of $App^{h/h}$ rats were studied longitudinally, A and B, housed across four Intelli-Cages separated by sex and cohort. Twelve rats were designated for each IntelliCage such that there would be 24 rats per cohort consisting of 12 females and 12 males each. The cohorts were run on separate program timelines, once at 6–8 weeks of age and again at 4–5 months of age (*Run-1* and *Run-2* through the program timeline, respectively), as outlined in Table 2 with program schematics depicted in the figures showing the results.

**IntelliCage programs.** <u>Free adaptation</u> (both cohorts, 1 day)—The rats may drink water ad libitum and explore the IntelliCage, familiarizing themselves with its layout; all bottle access doors open in response to any corner visit. <u>Nosepoke adaptation</u> (both cohorts, 1 day)—The rats learn they must activate a nosepoke sensor to open a water access door at any corner for seven seconds; this nosepoke mechanic remains active for every program hereafter. <u>Time adaptation</u> (Cohort A: 4 days, Cohort B: 2 days)—The rats may only drink between 8pm and 11pm at any corner, a time window called the *drinking session*. <u>Single corner restriction</u> (Cohort B only, 2 days)—All rats must drink from a single correct corner with the other corners being neutral during the drinking session. The correct corner changes after ninety minutes, such that the rats can drink at one corner during the first half of the drinking session and must switch to the opposite corner during the second half. Over two days, the correct corner designation follows the path 1->3 (1st day), then 2->4 (2nd day), covering all corners. <u>Place learning</u> (Cohort A only, 3 days)—The rats may only drink during the drinking session at a corner assigned to each of them; these assigned corners are considered correct, and the non-

**Table 2. IntelliCage program timeline overview for cohorts A and B.**

| Day | Cohort A | Cohort B |
|---|---|---|
| 1 | Free adaptation | Free adaptation |
| 2 | Nosepoke adaptation | Nosepoke adaptation |
| 3 | Time adaptation | Time adaptation |
| 4 | | |
| 5 | | Single corner restriction |
| 6 | | |
| 7 | Place learning (PL) | Place learning with corner switch (CS) |
| 8 | | |
| 9 | | |
| 10 | Place reversal (PR) | |
| 11 | | Behavioral sequencing (BS) |
| 12 | | |
| 13 | Behavioral sequencing (BS) | |
| 14 | | |
| 15 | | |
| 16 | Serial reversal (SR) | Serial reversal (SR) |
| 17 | | |
| 18 | | |
| 19 | | |

assigned corners are considered incorrect. <u>Place learning with corner switch</u> (Cohort B only, 4 days)—Each rat is assigned an initial correct corner where it can drink during the drinking session, as in place learning, with the other corners being incorrect. Every 45 minutes, the correct corner designations are switched according to the cycle (1->3->4->2[->1]). If corner 2 were the initial correct corner, the cycle would be shifted over once (2->1->3->4[->2]). After the first switch, the positions of the incorrect corners adjust accordingly. By the first 45-minute block of the next drinking session, the correct corner will have returned to its initial location. A *phase* refers to a 45-minute block during the drinking session in this program. The end of a phase marks when a corner switch occurs. <u>Place reversal</u> (Cohort A only, 3 days)—The rats may only drink during the drinking session at the corner diagonally opposite the one assigned in place learning; those reversed corners are considered correct, and the remaining corners, including the original assigned corner, are considered incorrect. <u>Behavioral sequencing</u> (Cohort A: 3 days, Cohort B: 5 days)—The rats must alternate between drinking at the initial learned corner and the opposite corner during the drinking session, so that one corner in the assigned diagonal is active (correct) at a time while the other is inactive (opposite); the conditions of the corners in the assigned diagonals alternate between correct and opposite, with a correct nosepoke triggering a condition switch. Visits to corners in the non-assigned diagonal are considered *lateral* visits. <u>Serial reversal</u> (both cohorts, 4 days)—The rats must alternate between a behavioral sequencing pattern on the original diagonal and the same on the other diagonal during the drinking session; the diagonal switches after every eight correct nosepokes. The corner conditions change as in the behavioral sequencing program, with lateral visits defined as before.

**Corner rank comparison.**   To understand better the effect of social interaction on the behavior of animals in the IntelliCage, we ranked animals via a point system based on whether an animal visits the correct corner more than other animals do during single corner restriction. This may occur with the exclusion of other animals from those corners, which may bear upon the performance of paired animals in subsequent tests. As the single corner restriction program changes the correct corner every ninety minutes over two drinking sessions, we followed this workflow to produce the rankings for each animal in cohort B: 1. For each 90-minute block of the program during the drinking session, rank the animals within each IntelliCage according to the number of visits made to the appropriate correct corner; there should be four lists for each IntelliCage, corresponding to the 8:00–9:30pm and 9:30–11:00pm periods of drinking sessions 1 and 2. 2. An animal is said to *out-visit* another animal at a corner if it makes more visits than the other one during a given time interval. Assign four scores to each animal equal to the number of animals it out-visits, one score for each 90-minute block; each corner should be represented once as a correct corner. For example, during the period when corner 1 is considered correct, if an animal visits corner 1 more times than 4 other animals, it receives a score of 4 for that 90-minute block. 3. Use the four scores generated for each animal per Run to calculate mean scores and standard errors for statistical analyses: we compared the mean scores of animals within each IntelliCage for a given Run, performing a one-way ordinary ANOVA in Prism 9 (GraphPad, San Diego, California) followed by Tukey's multiple comparisons tests when applicable ($p < 0.05$ was significant).

**Learning curves.**   To visualize learning of all the animals in each IntelliCage as a unit, we charted the fractional accumulation of correct visits (also opposite visits for the Behavioral sequencing and Serial reversal tasks) over the course of each drinking session. With the resulting curves, we can qualitatively compare task performance according to drinking session, sex, and Run. We followed this workflow to produce the learning curves for each program: 1. For each subset of rats by sex, cohort, and Run (e.g., cohort A males in their Run-1), count the total number of visits those rats made for each drinking session. For a 3-day program, there

should be 3 totals for a given subset. 2. For each subset as described in step 1, tabulate the fractional accumulation of correct (and opposite) visits over time for each drinking session, adding to each fraction, starting from 0, the value of 1 divided by the associated total count for that subset and drinking session each time a correct (or opposite) visit occurs, and 0 otherwise. The sum should be reset to 0 whenever a new drinking session occurs. 3. Match each fraction with a timestamp relative to the start of the first drinking session of a program, excluding time not belonging to a drinking session, e.g., if the $n$th fraction is associated with a visit that occurred during the 35th minute of the third drinking session of a program, the fraction is matched with minute 395 (180 + 180 + 35). 4. Plot the resulting tables with time as the independent variable and fraction as the dependent variable, yielding the learning curves.

**Inclusion/exclusion criteria.** During drinking sessions, animals may not visit or visit infrequently corners. Animals that did not make more than 25 visits during a drinking session in a learning and memory task were excluded from the analysis of that session. Animals that did not make sufficient visits for two consecutive drinking sessions were removed from the IntelliCage the following morning and allowed to drink water freely for an hour before being returned to the IntelliCage. Animals that died at any point during the timeline were excluded from the analyses of the current drinking session and, for obvious reasons, from all subsequent drinking sessions. Two cohort A females were excluded before the start of Run-2 because one died and the other developed hind limb paralysis. Two cohort B females died before the start of Run-1. No cohort A males were excluded from Run-1 analysis. No drinking session data points were excluded due to insufficient visits for any females of either cohort for either Run except for one during the 3rd day of place learning, cohort A, Run-1. One cohort A male was effectively excluded from all Run-2 analysis due to its insufficient visits throughout the timeline. One cohort B male died before the start of Run-2, and another died during the first day of behavioral sequencing during Run-2. One cohort B male was mostly excluded from Run-1 analysis of behavioral sequencing and serial reversal due to insufficient visits. No other data points from cohort B males had to be excluded during Run-2 analysis. Details of data point exclusion by drinking session can be seen in the tables accompanying data figures.

**Statistical analysis of area under the curves (AUC).** To assess task performance quantitatively, we used the area under the learning curves of individual animals in each IntelliCage. Every correct visit during a drinking session contributes to this area; this approach accounts not only for the total fraction of correct visits but also for the rate at which they accumulate. It also takes advantage of the large volume of data the IntelliCage collects from each program such that there is no need to approximate the rate of learning with curve fitting. For the Behavioral sequencing and the Serial reversal tasks we also calculated the AUC for the opposite corner, since the opposite corner represents the previously correct corner. Thus, calculations of these two areas indicates the speed by which a rat learns the rules regulating alternation of correct to previously correct corners. We followed this workflow for statistical analysis of learning for each program: 1. Tabulate the fractional accumulation for individual rats as described above; in other words, make the calculations necessary to generate learning curves for each rat in the IntelliCage rather than a group of them for every drinking session. 2. Calculate the area underneath each learning curve, bounded on the left and right by the start and end of each drinking session, respectively, and below by the x-axis. Before calculating the area, we completed the curve by extending it horizontally such that the final fraction at the end of the drinking session is equal to the fraction accumulated by the last visit the animal made during that drinking session. Each result is a data point representing the cumulative correct/opposite visit learning of a specific rat for a given drinking session. 3. Run statistical tests with those data points based on desired comparisons. We focused on three factors in our analysis: drinking session, sex, and Run. Given a Run and program, we performed a two-way repeated measures

ANOVA in Prism 9 on the data from all animals in a cohort, organized by sex and drinking session, followed by Šídák's multiple comparisons tests when applicable ($p < 0.05$ was significant). Specifically, we wanted to see whether there were significant session-wise differences within sex or sex differences within a given drinking session. If one or more drinking session data points were excluded for a given Run and program, mixed-effects analysis was performed instead with appropriate post-hoc tests. Paired t tests were performed to compare performance between Runs within a cohort for a given program, sex, and drinking session.

## Results

### $App^{h/h}$ rats produce human Aβ species and do not develop AD-like pathology up to 14 months of age

We have previously reported that the humanizing mutations do not alter APP expression levels [13]. To verify that rats used in these experiments contain the humanizing mutations in *App* exon-16, we amplified by PCR the *App* gene exon-16 from $App^{w/w}$ and $App^{h/h}$ rats. Sequencing of the PCR products shows that the humanizing mutations were correctly inserted in the $App^{h/h}$ genome (Fig 2A). Next, we verified whether $App^h$ rats produce human Aβ40 and Aβ42, and whether the protein products contain the humanizing mutations. Levels of human and rat Aβ40 and Aβ42 were measured in $App^{w/w}$ and $App^{h/h}$ rats' brains (5-week-old animals, 2 males and 2 females per genotype) using the species-specific detection antibodies 6E10 (human-specific) and M3.2 (rat/mouse-specific). As shown in Fig 2B, $App^{h/h}$ rats produce human Aβ species, while $App^{w/w}$ rats produce rat Aβ species. To complete our characterization, total brain lysates were analyzed by Western blot using the rat/mouse-specific M3.2 and the human-specific 6E10 antibodies. M3.2 detected APP only in $App^{w/w}$ brains, while 6E10 APP detected APP

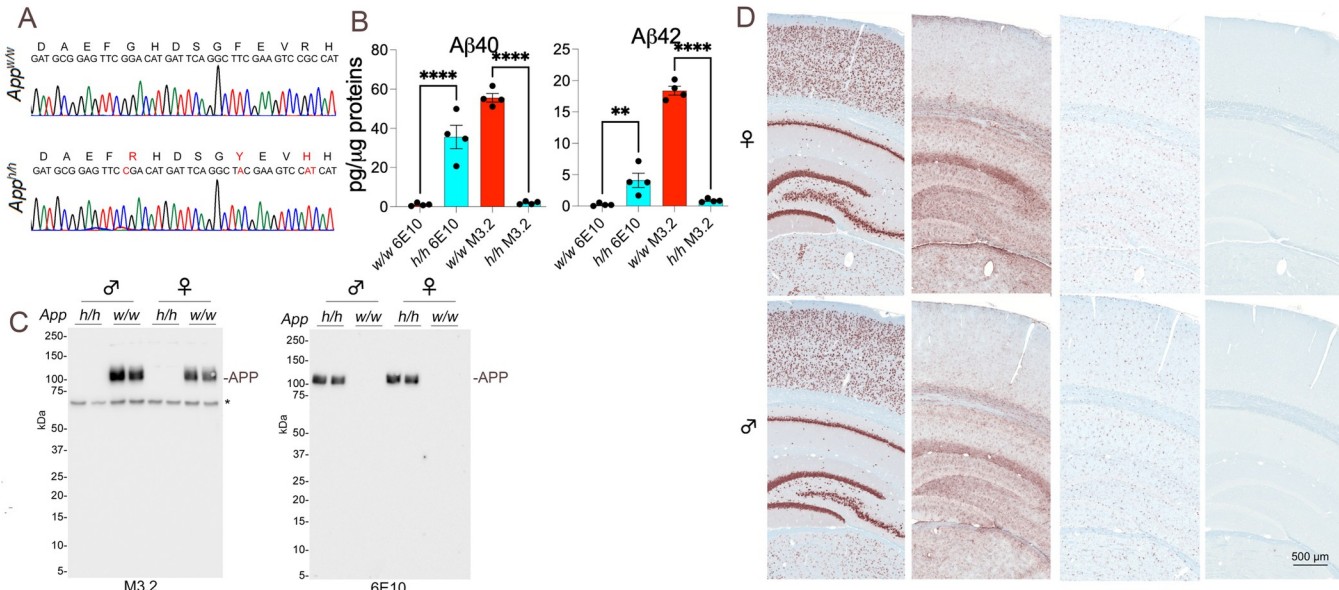

**Fig 2. Characterization of $App^{h/h}$ rats.** (**A**) To verify that the humanizing mutations were correctly inserted into *App* exon-16, exon-16 was amplified by PCR from $App^{w/w}$ and $App^{h/h}$ rats. PCR products' sequencing showed correct insertion of the humanizing mutations into the $App^{h/h}$ genome. Nucleotides (G to C, T to A, and GC to AT) and amino acids (G to R, Y to F, and R to H) substitutions are in red. (**B**) The human Aβ-region specific antibody 6E10 detects Aβ40 and Aβ42 in $App^{h/h}$ but not $App^{w/w}$ rats' brains. Conversely, the rat/mouse Aβ-region specific antibody M3.2 detects Aβ40 and Aβ42 in $App^{w/w}$ but not $App^{h/h}$ rats' brains. (**C**) Western blot analysis of brain lysates shows that 6E10 detects APP in $App^{h/h}$ but not $App^{w/w}$ rats. Conversely, M3.2 detects APP in $App^{w/w}$ but not $App^{h/h}$ rats. (**D**) $App^{h/h}$ rats do not show AD-like histopathology at 14 months of age. Six male and six females rats were studied. The figure shows representative IHC in a representative male and female subject. The AT8 samples are not shown because no signal was detected.

only in $App^{h/h}$ brains, confirming that APP produced by the $App^h$ alleles contains the humanized Aβ mutations (Fig 2C).

Finally, we tested whether expression of human Aβ species is sufficient to cause AD-like pathology in rats. For this, we performed histological and immunohistochemical (IHC) analyses on 14-month-old male and female $App^{h/h}$ rats. Staining with 6E10 was used to detect amyloid pathology; anti-NeuN was used to assess neuronal density and neurodegeneration; anti-IBA-1 was used to assess the activation state of microglia cells; anti-GFAP was used to assess the activation state of astrocytes; antibody AT8 was used to assess tau phosphorylation and neuronal tangle inclusion. Anti-NeuN staining did not show obvious neurodegeneration. The staining with anti-IBA-1 and anti-GFAP showed neither microglial or astrocytic activation, nor the presence of inflammatory foci and neuroinflammation. 6E10 and AT8 staining did not reveal amyloid plaques or tau pathology, respectively (Fig 2D). These results indicate that expression of human Aβ species *per se'*, is insufficient to prompt obvious AD-like pathology in rats (at least in 14-month-old rats). Therefore, $App^{h/h}$ rats represent a valid control group when the pathogenic effects of FAD and ADRD mutations, and of LOAD gene variants, are assessed.

## *App*<sup>h/h</sup> rats do not visit corners more often than other *App*<sup>h/h</sup> rats during single corner restriction in cohort B at either 6–8 weeks or 4–5 months of age

After IntelliCage adaptation as outlined in Table 2, rats in cohort B were started on the single corner restriction program, which tested whether the animals were able to share this corner equally among themselves for water (Fig 3A). The animals were assigned a rank during each 90-minute block of the two drinking sessions (four ranks total) based on visits to the actively correct corner, as described in the methods. The mean rank was used to compare animal learning (Fig 3B). There were no significant differences among male rats during either Run. During Run-1 one female rat (Animal 22) had a mean rank significantly lower than that of two other female rats (Animals 16 and 17), and Animal 14 had a significantly lower mean rank than that of Animal 16, but these differences were not observed during Run-2 for the same animals.

## *App*<sup>h/h</sup> rats in cohort A can acquire a place learning task by 6–8 weeks of age with session-wise improvement

After IntelliCage adaptation as outlined in Table 2, rats in cohort A were started on the place learning program (Fig 4A). Animal learning in this program and subsequent programs was summarized via (1) learning curves showing the fractional accumulation of correct/opposite corner visits of all animals during each drinking session, organized by sex and Run; and (2) comparisons of mean area under the learning curves of individual animals for correct/opposite visits during each drinking session, to quantify differences in task performance. Analysis of area under the curves (AUC) revealed significant session-wise increases for correct visits (C-AUC) for both sexes during Run-1. During Run-2, there were no significant session-wise differences in C-AUC for either sex (Fig 4B). There were no significant sex differences for either Run (Fig 4C), but C-AUC was significantly higher during Run-2 for all drinking sessions within females (Fig 4D). Qualitatively, a learning curve for correct visits that steepens session-wise signifies task acquisition (Fig 4E).

## *App*<sup>h/h</sup> rats in cohort A can acquire a place reversal task by 6–8 weeks of age with session-wise improvement

After place learning, rats in cohort A were started on the place reversal program, which switches the correct corner in place learning to the one diagonally opposing it (Fig 5A). There

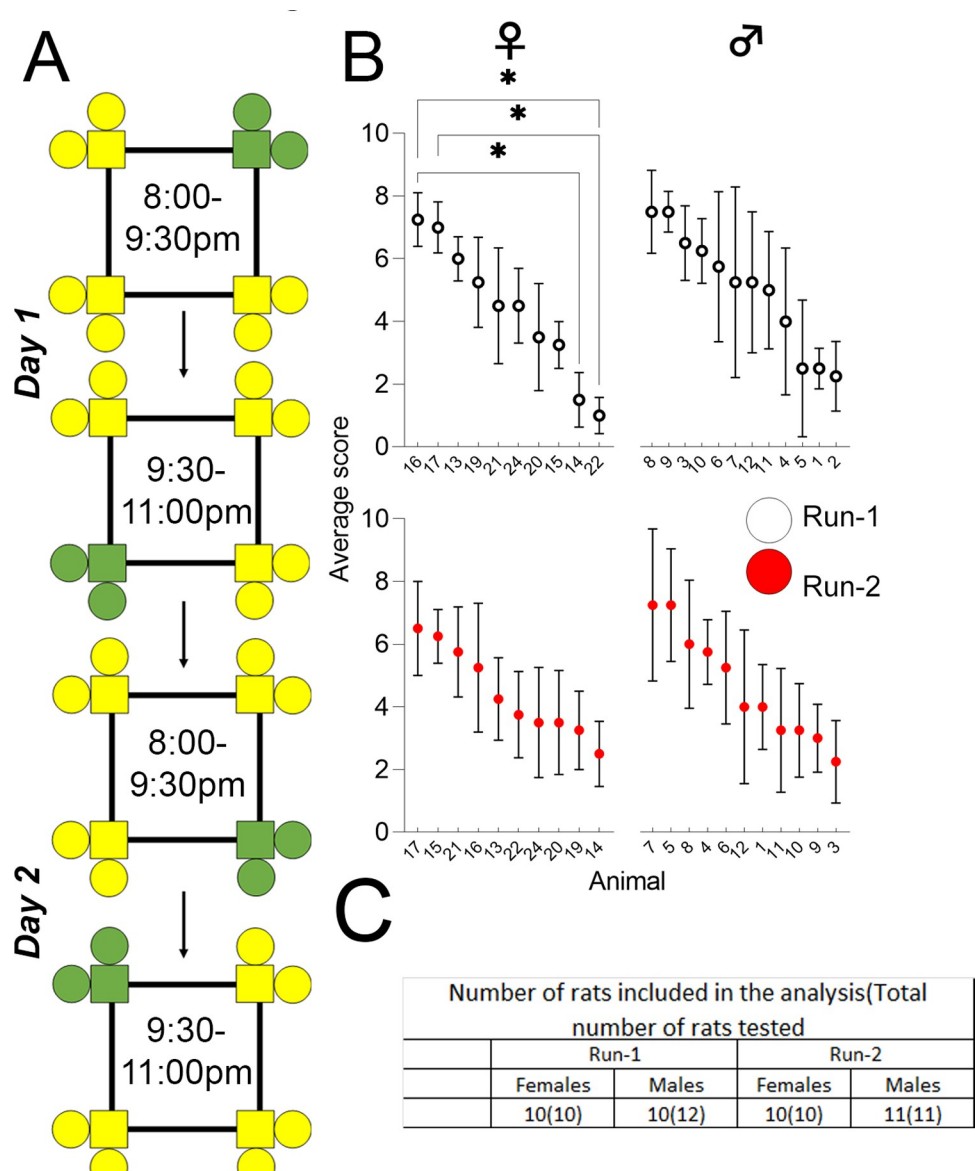

**Fig 3. Single corner restriction, cohort B. (A)** Single corner restriction (Cohort B only, 2 days). Progression of drinking (green) and non-drinking (yellow) corner layouts over two days. **(B)** Average scores for individual animals in cohort B by sex and Run, in decreasing order from left to right. Data points from Run-1 and Run-2 are indicated by white and red circles, respectively. All data represented as mean ± SEM (*$p < 0.05$). See Table 3 for statistical analysis. ♀ = female, ♂ = male. $n$ = 10 for females in both Runs. $n$ = 12 for males in Run-1, while $n$ = 11 for males in Run-2 due to death of one animal between Runs.

were also significant session-wise increases in C-AUC for both sexes during Run-1, with no significant differences seen during Run-2 for either sex (Fig 5B). There were significant sex differences seen during Run-1, with C-AUC higher for males for all drinking sessions, but not during Run-2 (Fig 5C). For females, C-AUC was significantly higher during Run-2 compared to Run-1 for the 1st and 3rd drinking sessions, with the value for the 2nd drinking session being higher too but not reaching significance; there were no significant differences between runs for males (Fig 5D). Learning curves showed qualitative improvement, like those shown for place learning (Figs 4E and 5E).

**Table 3. Statistical analysis of data shown in Fig 3 for single corner restriction, cohort B.**

| | Ordinary one-way ANOVA | | |
|---|---|---|---|
| **Female** | **Run** | **F (DFn, DFd)** | **P** |
| | Run-1 | F (9, 30) = 3.352 | 0.0060 |
| | Run-2 | F (9, 30) = 0.9019 | 0.5359 |
| | **post-hoc Tukey's multiple comp. test** | **Summary** | **Adjusted P** |
| | 14 (Run-1) vs. 16 (Run-1) | * | 0.0390 |
| | 16 (Run-1) vs. 22 (Run-1) | * | 0.0186 |
| | 17 (Run-1) vs. 22 (Run-1) | * | 0.0270 |
| **Male** | **Run** | **F (DFn, DFd)** | **P** |
| | Run-1 | F (11, 36) = 1.036 | 0.4366 |
| | Run-2 | F (10, 33) = 0.9633 | 0.4921 |

A *p*-value less than 0.05 is considered significant (*$p < 0.05$).

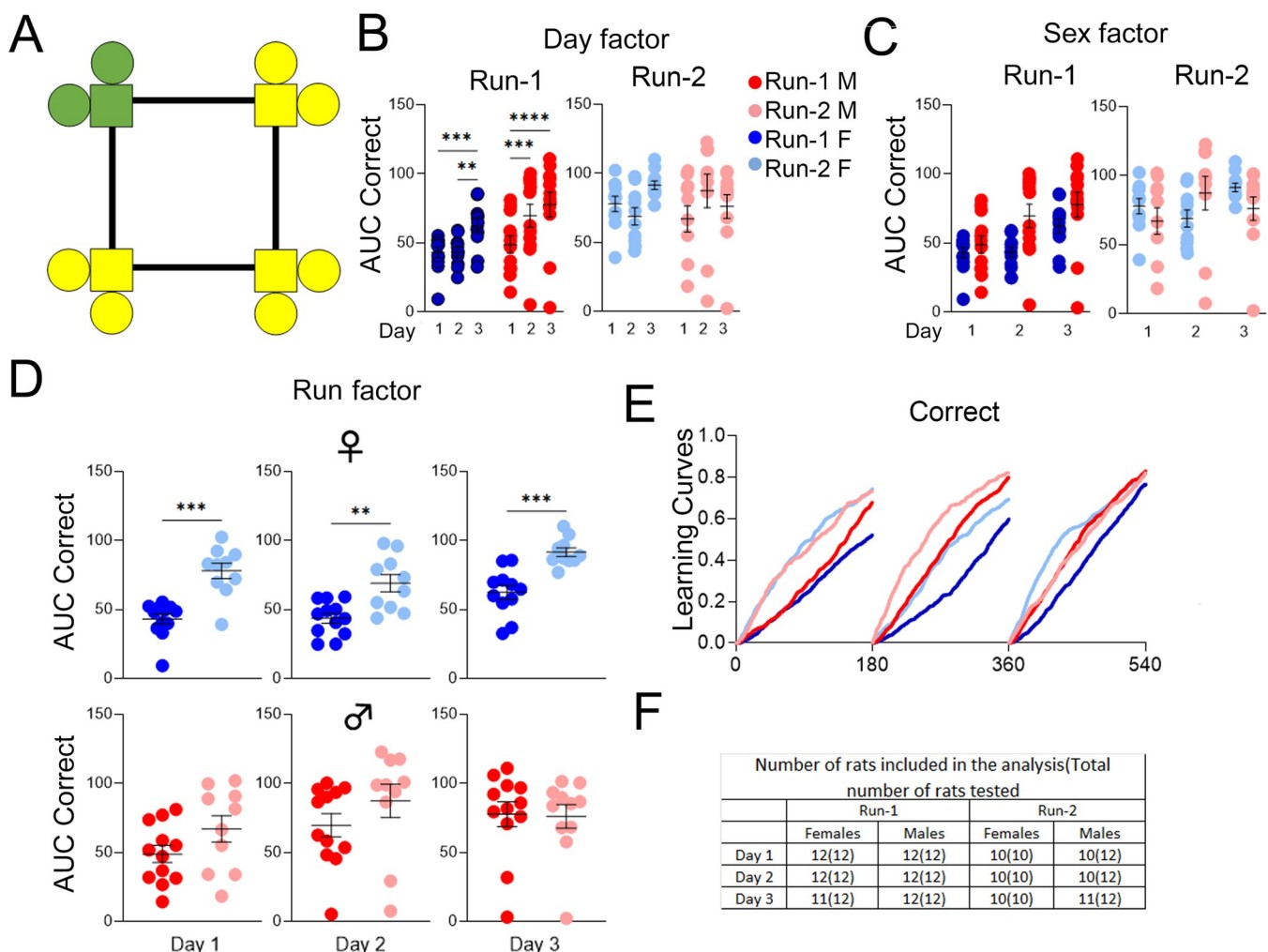

**Fig 4. Place learning, cohort A. (A)** Example of correct (green) and incorrect (yellow) corner layout for a rat assigned to corner 4. **(B)** Day factor, **(C)** Sex factor and **(D)** Run factor comparisons of area under the curve (AUC) for correct visit learning curves of individual animals by sex, program day, and Run. **(E)** Learning curves showing the fractional accumulation (y-axis) of correct visits over drinking session time (x-axis), reset every 180 minutes, by sex and Run. **(F)** For Run-1, 12 male and 12 female rats were tested. All 24 rats were included in the analysis of Run-1. For Run-2, 12 male and 10 female rats were tested, since two females died during the time between Runs. During Run-2, two males were excluded from the analysis of day 1 and 2, and one male was excluded from the analysis of day 3. These animals were excluded because they visited corners less than 25 times during the drinking session. All data represented as mean ± SEM (*$p < 0.05$, **$p < 0.01$, ***$p < 0.001$, ****$p < 0.0001$). See Tables 4 and 5 for statistical analysis. Males (M or ♂) are colored as red (Run-1) and light red (Run-2). Females (F or ♀) are colored as blue (Run-1) or light blue (Run-2).

Table 4. Statistical analysis of data shown in Fig 4B and 4C for place learning, cohort A, Run-1, and Run-2.

| Fig 4B and 4C | Mixed-effects analysis | | |
|---|---|---|---|
| Run-1 | Source of Variation | F (DFn, DFd) | P |
| | Interaction | F (2, 43) = 4.389 | 0.0184 |
| | Day factor(4B) | F (2, 43) = 24.08 | <0.0001 |
| | Sex factor(4C) | F (1, 22) = 3.847 | 0.0626 |
| | post-hoc Sidak's multiple comp. test | Summary | Adjusted P |
| | Female, day 1 vs. day 2 | ns | 0.9995 |
| | Female, day 1 vs. day 3 | *** | 0.0010 |
| | Female, day 2 vs. day 3 | ** | 0.0013 |
| | Male, day 1 vs. day 2 | *** | 0.0003 |
| | Male, day 1 vs. day 3 | **** | <0.0001 |
| | Male, day 2 vs. day 3 | ns | 0.2879 |
| Run-2 | Source of Variation | F (DFn, DFd) | P |
| | Interaction | F (2, 35) = 4.975 | 0.0125 |
| | Day factor(4B) | F (2, 35) = 2.064 | 0.1422 |
| | Sex factor(4C) | F (1, 20) = 0.2437 | 0.6269 |

A $p$-value less than 0.05 is considered significant ($^{**}p < 0.01$, $^{***}p < 0.001$, $^{****}p < 0.0001$). ns = not significant.

## $App^{h/h}$ rats in cohort B can acquire a place learning with corner switching task by 4–5 months of age with session-wise improvement

Rather than progressing from place learning to place reversal as cohort A rats did, cohort B rats were started on the place learning with corner switch program after single corner restriction. This program was designed to be a faster-paced combination of place learning and place reversal, with correct corners switching every 45 minutes within a drinking session (Fig 6A). No significant session-wise differences in C-AUC were seen during Run-1 for either sex, but there were significant increases in C-AUC during Run-2 for both sexes (Fig 6B). Sex differences were significant for all drinking sessions during Run-2, with C-AUC higher for males (Fig 6C). For females during Run-2 compared to Run-1, C-AUC was significantly higher for all but the 1st drinking session. For males during Run-2 compared to Run-1, C-AUC was significantly higher for all drinking sessions (Fig 6D). Learning curves qualitatively reflect this improvement across Runs for both sexes (Fig 6E).

Table 5. Statistical analysis of data shown in Fig 4D for place learning, cohort A, Run comparison.

| Fig 4D | Paired $t$ tests | | | |
|---|---|---|---|---|
| Run Factor | Comparison | t, df | Summary | P |
| | Female, day 1 | t = 5.672, df = 9 | *** | 0.0003 |
| | Male, day 1 | t = 1.141, df = 9 | ns | 0.2834 |
| | Female, day 2 | t = 4.655, df = 9 | ** | 0.0012 |
| | Male, day 2 | t = 0.9056, df = 9 | ns | 0.3888 |
| | Female, day 3 | t = 5.332, df = 9 | *** | 0.0005 |
| | Male, day 3 | t = 0.01567, df = 10 | ns | 0.9878 |

A $p$-value less than 0.05 is considered significant ($^{**}p < 0.01$, $^{***}p < 0.001$). ns = not significant.

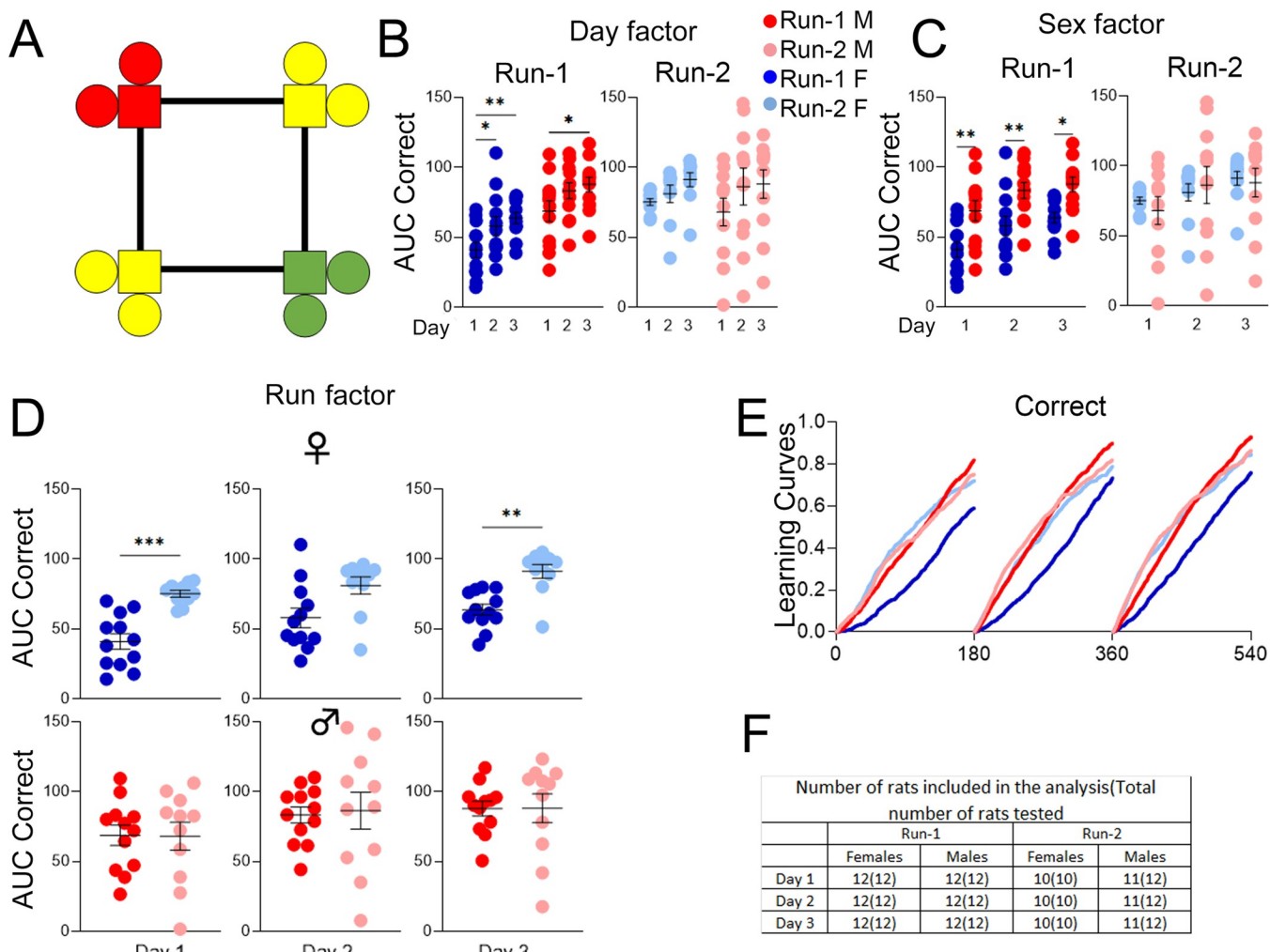

**Fig 5. Place reversal, cohort A. (A)** Example of correct (green) and incorrect (yellow, or red highlighting the previously correct corner) corner layout for a rat assigned to corner 4 during place learning. **(B)** Day factor, **(C)** Sex factor and **(D)** Run factor comparisons of area under the curve (AUC) for correct visit activity curves of individual animals by sex, program day, and Run. **(E)** Activity curves showing the fractional accumulation (y-axis) of correct visits over drinking session time (x-axis), reset every 180 minutes, by sex and Run. **(F)** For Run-1, 12 male and 12 female rats were tested. All 24 rats were included in the analysis of Run-1. For Run-2, 12 male and 10 female rats were tested, since two females died during the time between Runs. One male was excluded from Run-2 analysis because it visited corners less than 25 times during all 3 drinking sessions. All data represented as mean ± SEM (*$p < 0.05$, **$p < 0.01$, ***$p < 0.001$, ****$p < 0.0001$). See Tables 6 and 7 for statistical analysis. Males (M or ♂) are colored as red (Run-1) and light red (Run-2). Females (F or ♀) are colored as blue (Run-1) or light blue (Run-2).

## App^h/h rats in cohorts A and B can acquire a behavioral sequencing task by 6–8 weeks of age with session-wise improvement

After place learning and place reversal (cohort A) or place learning with corner switch (cohort B), we further tested the rats' spatial learning capabilities with a behavioral sequencing program requiring the animals to shuttle between diagonally opposing corners for water access (Fig 7A). Visits were categorized as correct (C), lateral, or opposite (O) as described in the methods, with learning curves generated (Figs 7E and 8D) and AUC analysis performed (Figs 7B–7D and 8A–7C) similarly as in other programs. Cohort A rats of both sexes during Run-1 showed significant increases in C-AUC and decreases in O-AUC. These changes were consistent during Run-2 for females, whereas males no longer showed significant session-wise

**Table 6. Statistical analysis of data shown in Fig 5B and 5C for place reversal, cohort A, Run-1, and Run-2.**

| Fig 5B and 5C | Two-way RM ANOVA | | |
|---|---|---|---|
| Run-1 | **Source of Variation** | **F (DFn, DFd)** | **P** |
| | Interaction | F (2, 44) = 0.08855 | 0.9154 |
| | Day factor(5B) | F (2, 44) = 12.26 | <0.0001 |
| | Sex factor(5C) | F (1, 22) = 15.38 | 0.0007 |
| | **post-hoc Sidak's multiple comp. test** | **Summary** | **Adjusted P** |
| | Female, day 1 vs. day 2 | * | 0.0277 |
| | Female, day 1 vs. day 3 | ** | 0.0020 |
| | Female, day 2 vs. day 3 | ns | 0.7307 |
| | Male, day 1 vs. day 2 | ns | 0.0713 |
| | Male, day 1 vs. day 3 | * | 0.0109 |
| | Male, day 2 vs. day 3 | ns | 0.8461 |
| | Female vs. Male, day 1 | ** | 0.0042 |
| | Female vs. Male, day 2 | ** | 0.0100 |
| | Female vs. Male, day 3 | * | 0.0153 |
| Run-12 | **Source of Variation** | **F (DFn, DFd)** | **P** |
| | Interaction | F (2, 38) = 0.4885 | 0.6174 |
| | Day factor(5B) | F (2, 38) = 4.223 | 0.0221 |
| | Sex factor(5C) | F (1, 19) = 0.02749 | 0.8701 |
| | **post-hoc Sidak's multiple comp. test** | **Summary** | **Adjusted P** |
| | Female, day 1 vs. day 2 | ns | 0.8887 |
| | Female, day 1 vs. day 3 | ns | 0.2396 |
| | Female, day 2 vs. day 3 | ns | 0.6212 |
| | Male, day 1 vs. day 2 | ns | 0.1257 |
| | Male, day 1 vs. day 3 | ns | 0.0804 |
| | Male, day 2 vs. day 3 | ns | 0.9957 |

A *p*-value less than 0.05 is considered significant (*$p < 0.05$, **$p < 0.01$). ns = not significant.

changes in C-AUC (Fig 7B). There were no significant sex differences observed for either Run (Fig 7C). For females during Run-2 compared to Run-1, C-AUC was significantly higher for the 1st drinking session; there were no other significant differences across Runs for any drinking session (Fig 7D).

Cohort B rats exhibited a different learning profile: for females, the session-wise differences in C-AUC and O-AUC were insignificant during both Runs, whereas for males, there were

**Table 7. Statistical analysis of data shown in Fig 5D for place reversal, cohort A, Run comparison.**

| Fig 5D | Paired t tests | | | |
|---|---|---|---|---|
| Run Factor | **Comparison** | **t, df** | **Summary** | **P** |
| | Female, day 1 | t = 5.188, df = 9 | *** | 0.0006 |
| | Male, day 1 | t = 0.06592, df = 10 | ns | 0.9487 |
| | Female, day 2 | t = 1.595, df = 9 | ns | 0.1453 |
| | Male, day 2 | t = 0.3032, df = 10 | ns | 0.7680 |
| | Female, day 3 | t = 3.329, df = 9 | ** | 0.0088 |
| | Male, day 3 | t = 0.09445, df = 10 | ns | 0.9266 |

A *p*-value less than 0.05 is considered significant (**$p < 0.01$, ***$p < 0.001$). ns = not significant.

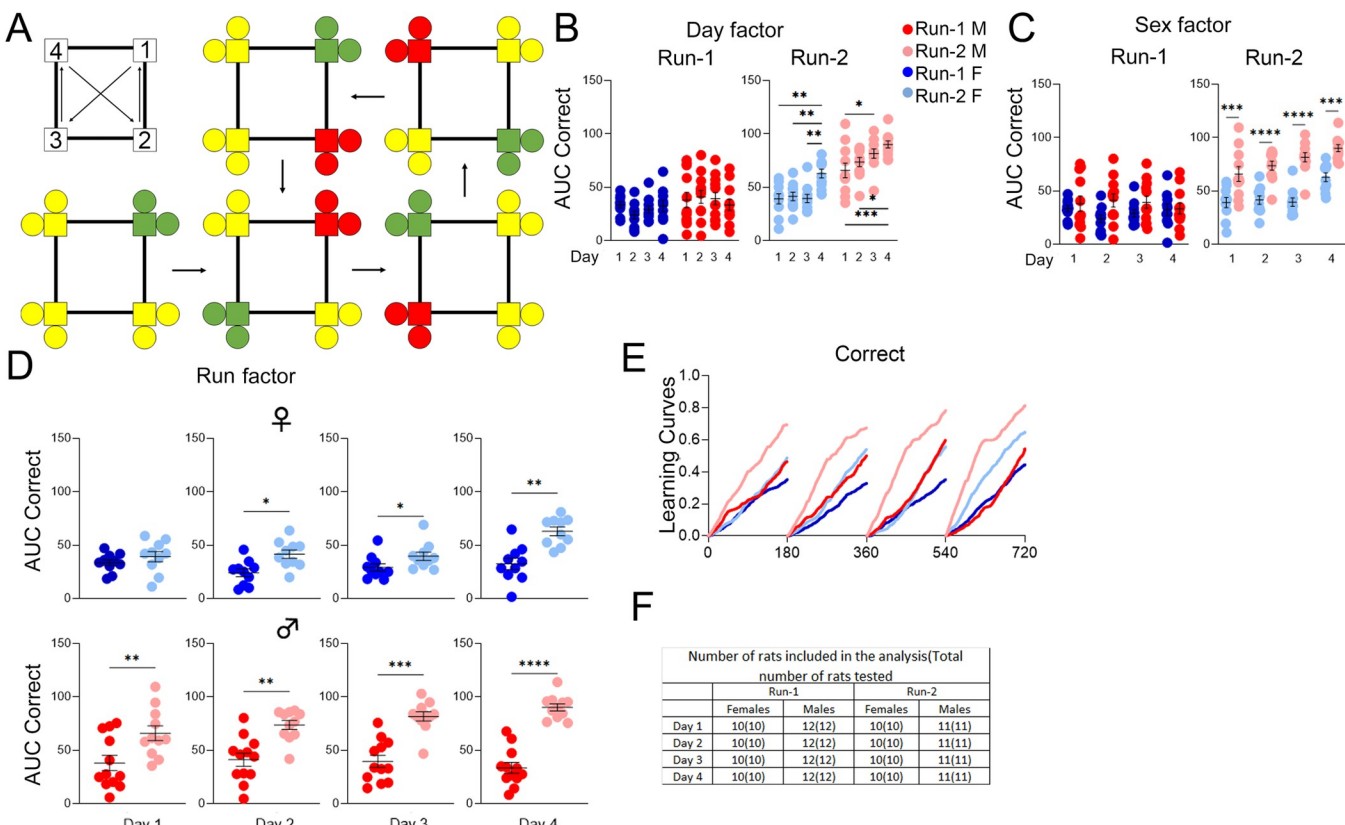

**Fig 6. Place learning with corner switch, cohort B. (A)** On the top left is the cycle of correct corners with movement every 45 minutes. The rest of the panel, starting from the bottom left, depicts an example of correct (green) and incorrect (yellow, or red highlighting the previously correct corner) layouts for a rat initially assigned to corner 1 and their cycle over the four phases of a drinking session, which ends with the top right layout before returning to the top center layout during the start of the next drinking session. **(B)** Day factor, **(C)** Sex factor and **(D)** Run factor comparisons of area under the curve (AUC) for correct visit activity curves of individual animals by sex, program day, and Run. **(E)** Activity curves showing the fractional accumulation (y-axis) of correct visits over drinking session time (x-axis), reset every 180 minutes, by sex and Run. **(F)** For Run-1, 12 male and 10 female rats were tested since 2 females died just before the timeline started. All 22 rats were included in the analysis of Run-1. For Run-2, 11 male and 10 female rats were tested, since one male died during the time between Runs. All 21 rats were included in the analysis of Run-2. All data represented as mean ± SEM (*$p < 0.05$, **$p < 0.01$, ***$p < 0.001$, ****$p < 0.0001$,). See Tables 8 and 9 for statistical analysis. Males (M or ♂) are colored as red (Run-1) and light red (Run-2). Females (F or ♀) are colored as blue (Run-1) or light blue (Run-2).

many significant session-wise differences, especially during Run-2 with increases in C-AUC and decreases in O-AUC (Fig 8A). Sex differences were insignificant during Run-1, whereas C-AUC was significantly higher in males during Run-2 from the 2nd to the 5th drinking sessions (Fig 8B). Significant differences found between Runs for females were sporadic for C-AUC (1st drinking session, higher in Run-1) and O-AUC (1st and 3rd drinking sessions, higher in Run-2); in contrast, for males, C-AUC was significantly higher for every drinking session during Run-2 compared to Run-1, with O-AUC higher for the 1st and 2nd drinking sessions (Fig 8C).

### *App*^h/h rats in cohort A and B may not be able to acquire a serial reversal task by 4–5 months of age

We ended the timeline for both cohorts with a serial reversal program designed to add a layer of complexity to behavioral sequencing by requiring the rats to alternate diagonals after every eight successive, but not necessarily consecutive, correct nosepokes correct nosepokes (Fig 9A). For cohort A, qualitatively, learning curves for both sexes did not show much difference

**Table 8. Statistical analysis of data shown in Fig 6B and 6C for place learning with corner switch, cohort B, Run-1, and Run-2.**

| Fig 6B and 6C | Two-way RM ANOVA | | |
|---|---|---|---|
| Run-1 | **Source of Variation** | **F (DFn, DFd)** | **P** |
| | Interaction | F (3, 60) = 1.425 | 0.2444 |
| | Day factor(6B) | F (3, 60) = 0.2456 | 0.8639 |
| | Sex factor(6C) | F (1, 20) = 2.212 | 0.1526 |
| Run-2 | **Source of Variation** | **F (DFn, DFd)** | **P** |
| | Interaction | F (3, 57) = 1.510 | 0.2217 |
| | Day factor(6B) | F (3, 57) = 12.64 | <0.0001 |
| | Sex factor(6C) | F (1, 19) = 61.50 | <0.0001 |
| | **post-hoc Sidak's multiple comp. test** | **Summary** | **Adjusted P** |
| | Female, day 1 vs. day 2 | ns | 0.9993 |
| | Female, day 1 vs. day 3 | ns | >0.9999 |
| | Female, day 1 vs. day 4 | ** | 0.0012 |
| | Female, day 2 vs. day 3 | ns | 0.9997 |
| | Female, day 2 vs. day 4 | ** | 0.0042 |
| | Female, day 3 vs. day 4 | ** | 0.0014 |
| | Male, day 1 vs. day 2 | ns | 0.6828 |
| | Male, day 1 vs. day 3 | * | 0.0434 |
| | Male, day 1 vs. day 4 | *** | 0.0004 |
| | Male, day 2 vs. day 3 | ns | 0.6633 |
| | Male, day 2 vs. day 4 | * | 0.0315 |
| | Male, day 3 vs. day 4 | ns | 0.5969 |
| | Female vs. Male, day 1 | *** | 0.0004 |
| | Female vs. Male, day 2 | **** | <0.0001 |
| | Female vs. Male, day 3 | **** | <0.0001 |
| | Female vs. Male, day 4 | *** | 0.0003 |

A $p$-value less than 0.05 is considered significant (*$p < 0.05$, **$p < 0.01$, ***$p < 0.001$,****$p < 0.0001$). ns = not significant.

between Runs or session-wise improvement (Fig 9E). Session-wise differences in AUC were minimal for both sexes during both Runs (Fig 9B). Sex differences were insignificant for both Runs as well (Fig 9C). The only significant difference between Runs was in O-AUC for females

**Table 9. Statistical analysis of data shown in Fig 6D for place learning with corner switch, cohort B, Run comparison.**

| Fig 6D | Paired $t$ tests | | | |
|---|---|---|---|---|
| Run Factor | **Comparison** | **t, df** | **Summary** | **P** |
| | Female, day 1 | t = 1.200, df = 9 | ns | 0.2606 |
| | Male, day 1 | t = 3.216, df = 10 | ** | 0.0092 |
| | Female, day 2 | t = 2.577, df = 9 | * | 0.0298 |
| | Male, day 2 | t = 4.572, df = 10 | ** | 0.0010 |
| | Female, day 3 | t = 3.192, df = 9 | * | 0.0110 |
| | Male, day 3 | t = 5.835, df = 10 | *** | 0.0002 |
| | Female, day 4 | t = 4.269, df = 9 | ** | 0.0021 |
| | Male, day 4 | t = 8.317, df = 10 | **** | <0.0001 |

A $p$-value less than 0.05 is considered significant (*$p < 0.05$, **$p < 0.01$, ***$p < 0.001$, ****$p < 0.0001$). ns = not significant.

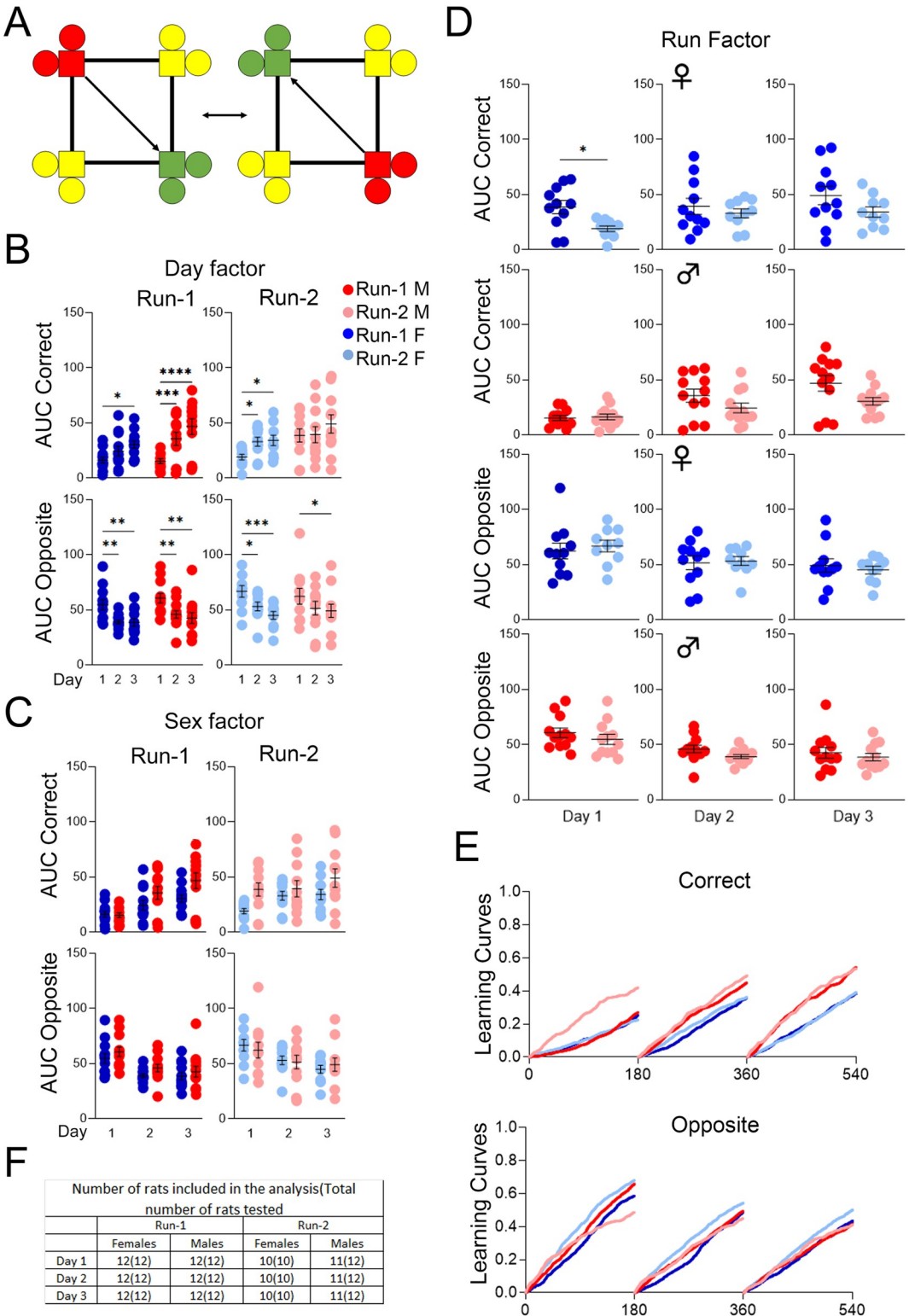

**Fig 7. Behavioral sequencing, cohort A. (A)** Example of correct (green), lateral (yellow), and opposite (red) corner layouts and pattern for a rat initially assigned to either corner 2 or 4. **(B)** Day factor, **(C)** Sex factor and **(D)** Run factor comparisons of area under the curve (AUC) for activity curves of individual animals by sex, program day, and Run, separated by visit categories (Correct or Opposite). **(E)** Activity curves showing the fractional accumulation (y-axis) of correct visits or opposite visits over drinking session time (x-axis), reset every 180 minutes, by sex and Run. **(F)** For Run-1, 12 male and 12 female rats

were tested. All 24 rats were included in the analysis of Run-1. For Run-2, 12 male and 10 female rats were tested, since two females died during the time between Runs. One male was excluded from Run-2 analysis because it visited corners less than 25 times during all 3 drinking sessions. All data represented as mean ± SEM (*$p < 0.05$, ** $p < 0.01$, *** $p < 0.001$, **** $p < 0.0001$). See Tables 10 and 11 for statistical analysis. Males (M or ♂) are colored as red (Run-1) and light red (Run-2). Females (F or ♀) are colored as blue (Run-1) or light blue (Run-2).

during the 1st drinking session, which was lower in Run-2 (Fig 10C). For cohort B, the learning curves suggest a possible difference between Run-1 and Run-2 for males, but no session-wise differences (Fig 10D). AUC analysis revealed no significant session-wise differences (Fig 10A). Compared to females during Run-2, C-AUC was significantly higher for all drinking sessions in males, with O-AUC significantly lower for the 3rd and 4th drinking sessions (Fig 10B). For males during Run-2 compared to Run-1, C-AUC was significantly higher for every drinking

**Table 10. Statistical analysis of data shown in Fig 7B and 7C for behavioral sequencing, cohort A, Run-1, and Run-2.**

| Fig 7B and 7C | Two-way RM ANOVA | | |
|---|---|---|---|
| **Correct Run-1** | **Source of Variation** | **F (DFn, DFd)** | **P** |
| | Interaction | F (2, 44) = 3.323 | 0.0453 |
| | Day factor(7B) | F (2, 44) = 22.41 | <0.0001 |
| | Sex factor(7C) | F (1, 22) = 2.906 | 0.1023 |
| | **post-hoc Sidak's multiple comp. test** | **Summary** | **Adjusted P** |
| | Female, day 1 vs. day 2 | ns | 0.2721 |
| | Female, day 1 vs. day 3 | * | 0.0168 |
| | Female, day 2 vs. day 3 | ns | 0.5330 |
| | Male, day 1 vs. day 2 | *** | 0.0004 |
| | Male, day 1 vs. day 3 | **** | <0.0001 |
| | Male, day 2 vs. day 3 | ns | 0.0775 |
| **Correct Run-2** | **Source of Variation** | **F (DFn, DFd)** | **P** |
| | Interaction | F (2, 38) = 1.537 | 0.2281 |
| | Day factor(7B) | F (2, 38) = 5.618 | 0.0073 |
| | Sex factor(7C) | F (1, 19) = 3.569 | 0.0742 |
| | **post-hoc Sidak's multiple comp. test** | **Summary** | **Adjusted P** |
| | Female, day 1 vs. day 2 | * | 0.0474 |
| | Female, day 1 vs. day 3 | * | 0.0278 |
| | Female, day 2 vs. day 3 | ns | 0.9950 |
| | Male, day 1 vs. day 2 | ns | 0.9988 |
| | Male, day 1 vs. day 3 | ns | 0.1598 |
| | Male, day 2 vs. day 3 | ns | 0.2075 |
| **Opposite Run-1** | **Source of Variation** | **F (DFn, DFd)** | **P** |
| | Interaction | F (2, 44) = 0.09638 | 0.9083 |
| | Day factor(7B) | F (2, 44) = 15.98 | <0.0001 |
| | Sex factor(7C) | F (1, 22) = 2.116 | 0.1599 |
| | **post-hoc Sidak's multiple comp. test** | **Summary** | **Adjusted P** |
| | Female, day 1 vs. day 2 | ** | 0.0055 |
| | Female, day 1 vs. day 3 | ** | 0.0039 |
| | Female, day 2 vs. day 3 | ns | 0.9992 |
| | Male, day 1 vs. day 2 | ** | 0.0088 |
| | Male, day 1 vs. day 3 | ** | 0.0011 |
| | Male, day 2 vs. day 3 | ns | 0.8555 |

(*Continued*)

**Table 10.** (Continued)

| Fig 7B and 7C | Two-way RM ANOVA | | |
|---|---|---|---|
| Opposite Run-2 | **Source of Variation** | **F (DFn, DFd)** | **P** |
| | Interaction | F (2, 38) = 0.7340 | 0.4867 |
| | Day factor(7B) | F (2, 38) = 12.12 | <0.0001 |
| | Sex factor(7C) | F (1, 19) = 0.009860 | 0.9219 |
| | **post-hoc Sidak's multiple comp. test** | **Summary** | **Adjusted P** |
| | Female, day 1 vs. day 2 | * | 0.0395 |
| | Female, day 1 vs. day 3 | *** | 0.0006 |
| | Female, day 2 vs. day 3 | ns | 0.3448 |
| | Male, day 1 vs. day 2 | ns | 0.1123 |
| | Male, day 1 vs. day 3 | * | 0.0379 |
| | Male, day 2 vs. day 3 | ns | 0.9526 |

A $p$-value less than 0.05 is considered significant (*$p < 0.05$, **$p < 0.01$, ***$p < 0.001$, ****$p < 0.0001$). ns = not significant.

session, and O-AUC was significantly lower for the 1st drinking session; for females, there were no significant differences between Runs (Fig 10C).

## Discussion

Place learning with reversal is the most frequently used learning protocol for the IntelliCage [36]. It has been applied to male Sprague-Dawley rats, also 6–8 weeks old, to explore the relationship between $GABA_B$ receptors and recognition memory [35]. In another study, an IntelliCage place learning task was performed alongside more traditional paradigms, namely, the forced swimming test, open field test, and Morris water maze, to validate a novel multi-function closed maze designed to detect learning, memory, and affective disorders in post-weaning socially isolated rats [40]. The results of the place learning task agreed with those of the Morris water maze with respect to spatial learning and memory deficits in socially isolated rats,

**Table 11. Statistical analysis of data shown in Fig 7D for behavioral sequencing, cohort A, Run comparison.**

| Fig 7D | Paired $t$ tests | | | |
|---|---|---|---|---|
| Run Factor Correct | **Comparison** | **t, df** | **Summary** | **P** |
| | Female, day 1 | t = 2.491, df = 8 | * | 0.0375 |
| | Male, day 1 | t = 0.2282, df = 11 | ns | 0.8237 |
| | Female, day 2 | t = 1.032, df = 8 | ns | 0.3324 |
| | Male, day 2 | t = 1.474, df = 11 | ns | 0.1686 |
| | Female, day 3 | t = 1.177, df = 8 | ns | 0.2728 |
| | Male, day 3 | t = 1.844, df = 11 | ns | 0.0923 |
| Run Factor Opposite | **Comparison** | **t, df** | **Summary** | **P** |
| | Female, day 1 | t = 0.9581, df = 8 | ns | 0.3661 |
| | Male, day 1 | t = 0.9490, df = 11 | ns | 0.3630 |
| | Female, day 2 | t = 0.6758, df = 8 | ns | 0.5182 |
| | Male, day 2 | t = 1.496, df = 11 | ns | 0.1628 |
| | Female, day 3 | t = 0.4350, df = 8 | ns | 0.6751 |
| | Male, day 3 | t = 0.7209, df = 11 | ns | 0.4860 |

A $p$-value less than 0.05 is considered significant (*$p < 0.05$). ns = not significant.

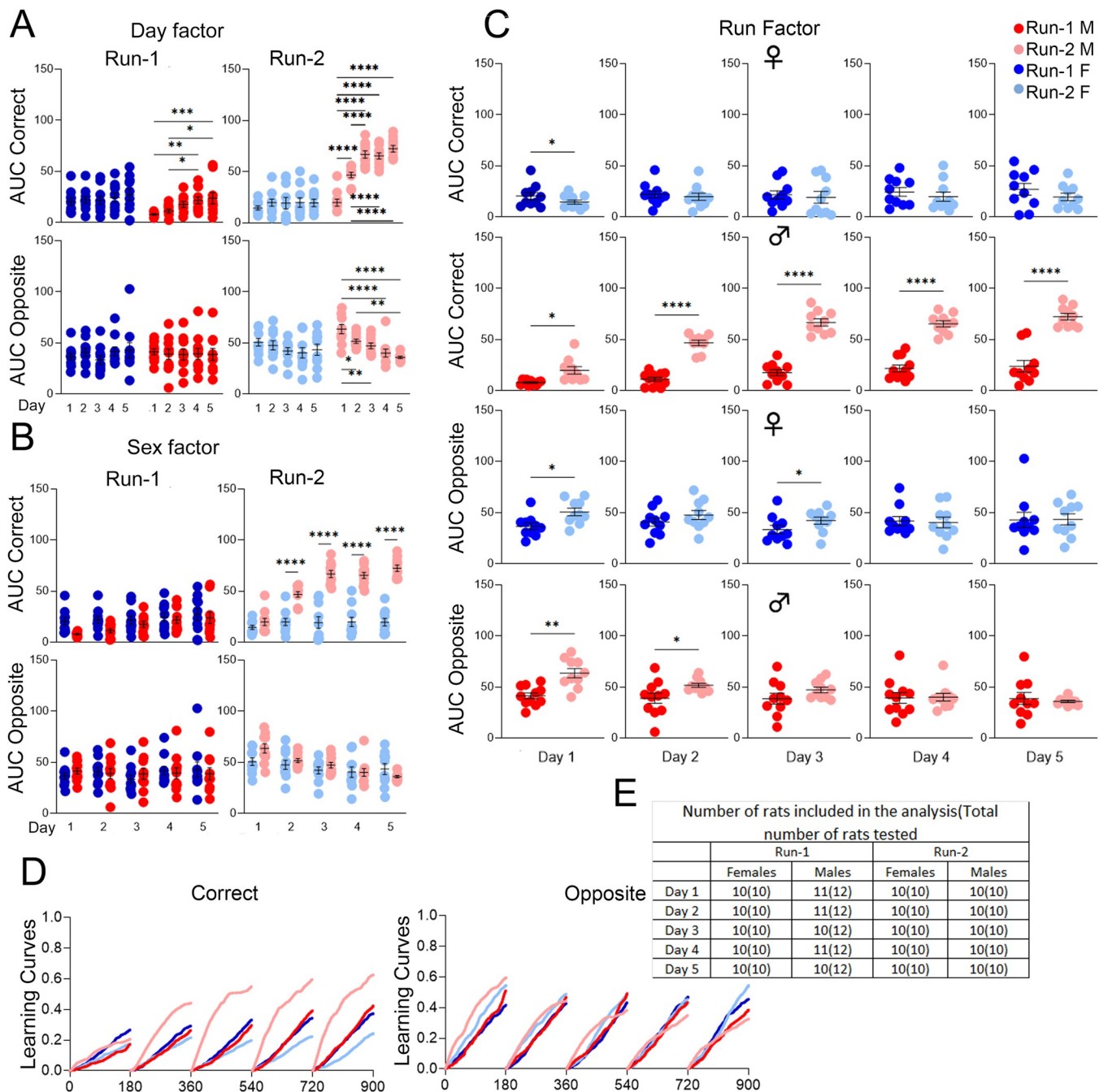

**Fig 8. Behavioral sequencing, cohort B. (A)** Day factor, **(B)** Sex factor and **(C)** Run factor comparisons of area under the curve (AUC) for activity curves of individual animals by sex, program day, and Run, separated by visit categories (Correct, BS-C or Opposite, BS-O). **(D)** Activity curves showing the fractional accumulation (y-axis) of correct visits or opposite visits over drinking session time (x-axis), reset every 180 minutes, by sex and Run. **(E)** For Run-1, 12 male and 10 female rats were tested since 2 females died just before the timeline started. One male was excluded from the analysis of days 1, 2, and 4, while two males were excluded from the analysis of days 3 and 5 during Run-2. These animals were excluded because they visited corners less than 25 times during those drinking sessions. For Run-2, 10 male and 10 female rats were tested, since one male died during the time between Runs, and one male died after place learning with corner switch. All 20 rats were included in the analysis of Run-2. All data represented as mean ± SEM (*$p < 0.05$, **$p < 0.01$, ***$p < 0.001$, ****$p < 0.0001$). See Tables 12 and 13 for statistical analysis. Males (M or ♂) are colored as red (Run-1) and light red (Run-2). Females (F or ♀) are colored as blue (Run-1) or light blue (Run-2).

**Table 12. Statistical analysis of data shown in Fig 8A and 8B for behavioral sequencing, cohort B, Run-1, and Run-2.**

| Fig 8A and 8B | Mixed-effects analysis | | |
|---|---|---|---|
| Correct Run-1 | **Source of Variation** | **F (DFn, DFd)** | **P** |
| | Interaction | F (4, 73) = 1.712 | 0.1565 |
| | Day factor(8A) | F (4, 73) = 6.442 | 0.0002 |
| | Sex factor(8B) | F (1, 20) = 2.837 | 0.1077 |
| | **post-hoc Sidak's multiple comp. test** | **Summary** | **Adjusted P** |
| | Male, day 1 vs. day 4 | ** | 0.0013 |
| | Male, day 1 vs. day 5 | *** | 0.0003 |
| | Male, day 2 vs. day 4 | * | 0.0367 |
| | Male, day 2 vs. day 5 | * | 0.0102 |
| Correct Run-2 | **Source of Variation** | **F (DFn, DFd)** | **P** |
| | Interaction | F (4, 72) = 30.28 | <0.0001 |
| | Day factor(8A) | F (4, 72) = 43.82 | <0.0001 |
| | Sex factor(8B) | F (1, 18) = 76.20 | <0.0001 |
| | **post-hoc Sidak's multiple comp. test** | **Summary** | **Adjusted P** |
| | Male, day 1 vs. day 2 | **** | <0.0001 |
| | Male, day 1 vs. day 3 | **** | <0.0001 |
| | Male, day 1 vs. day 4 | **** | <0.0001 |
| | Male, day 1 vs. day 5 | **** | <0.0001 |
| | Male, day 2 vs. day 3 | **** | <0.0001 |
| | Male, day 2 vs. day 4 | **** | <0.0001 |
| | Male, day 2 vs. day 5 | **** | <0.0001 |
| | Female vs. Male, day 2 | **** | <0.0001 |
| | Female vs. Male, day 3 | **** | <0.0001 |
| | Female vs. Male, day 4 | **** | <0.0001 |
| | Female vs. Male, day 5 | **** | <0.0001 |
| Opposite Run-1 | **Source of Variation** | **F (DFn, DFd)** | **P** |
| | Interaction | F (4, 73) = 0.5797 | 0.6783 |
| | Day factor(8A) | F (4, 73) = 0.5545 | 0.6963 |
| | Sex factor(8B) | F (1, 20) = 0.0003232 | 0.9858 |
| Opposite Run-2 | **Source of Variation** | **Summary** | **Adjusted P** |
| | Interaction | F (4, 72) = 3.377 | 0.0137 |
| | Day factor(8A) | F (4, 72) = 12.53 | <0.0001 |
| | Sex factor(8B) | F (1, 18) = 0.5148 | 0.4823 |
| | **post-hoc Sidak's multiple comp. test** | **Summary** | **Adjusted P** |
| | Male, day 1 vs. day 2 | * | 0.0490 |
| | Male, day 1 vs. day 3 | ** | 0.0016 |
| | Male, day 1 vs. day 4 | **** | <0.0001 |
| | Male, day 1 vs. day 5 | **** | <0.0001 |
| | Male, day 2 vs. day 5 | ** | 0.0025 |

A $p$-value less than 0.05 is considered significant ($^*p < 0.05$, $^{**}p < 0.01$, $^{***}p < 0.001$, $^{****}p < 0.0001$). ns not shown.

supporting the IntelliCage as a valid methodology beside traditional ones. In our study, $App^{h/h}$ rats of both sexes were able to adapt to the IntelliCage and acquire a place learning and reversal task, as well as a more complex behavioral sequencing task, by 6–8 weeks of age. Males tended to perform better than females at 4–5 months of age in place learning with corner switch—essentially a quicker version of the place learning with reversal paradigm—and behavioral

**Table 13. Statistical analysis of data shown in Fig 8C for behavioral sequencing, cohort B, Run comparison.**

| Fig 8C | Paired *t* tests | | | |
|---|---|---|---|---|
| Run Factor Correct | Comparison | t, df | Summary | P |
| | Female, day 1 | t = 2.393, df = 9 | * | 0.0403 |
| | Male, day 1 | t = 2.692, df = 8 | * | 0.0274 |
| | Female, day 2 | t = 0.8613, df = 9 | ns | 0.4114 |
| | Male, day 2 | t = 10.57, df = 8 | **** | <0.0001 |
| | Female, day 3 | t = 0.8399, df = 9 | ns | 0.4227 |
| | Male, day 3 | t = 10.76, df = 8 | **** | <0.0001 |
| | Female, day 4 | t = 1.226, df = 9 | ns | 0.2512 |
| | Male, day 4 | t = 8.341, df = 8 | **** | <0.0001 |
| | Female, day 5 | t = 1.809, df = 9 | ns | 0.1039 |
| | Male, day 5 | t = 7.820, df = 8 | **** | <0.0001 |
| Run Factor Opposite | Comparison | t, df | Summary | P |
| | Female, day 1 | t = 3.144, df = 9 | * | 0.0119 |
| | Male, day 1 | t = 3.395, df = 8 | ** | 0.0094 |
| | Female, day 2 | t = 1.403, df = 9 | ns | 0.1942 |
| | Male, day 2 | t = 3.111, df = 8 | * | 0.0144 |
| | Female, day 3 | t = 2.459, df = 9 | * | 0.0362 |
| | Male, day 3 | t = 1.404, df = 8 | ns | 0.1976 |
| | Female, day 4 | t = 0.4780, df = 9 | ns | 0.6441 |
| | Male, day 4 | t = 0.5876, df = 8 | ns | 0.5730 |
| | Female, day 5 | t = 0.08419, df = 9 | ns | 0.9347 |
| | Male, day 5 | t = 0.7311, df = 8 | ns | 0.4856 |

A *p*-value less than 0.05 is considered significant (*$p < 0.05$, **$p < 0.01$, ***$p < 0.001$, ****$p < 0.0001$). ns = not significant.

sequencing. This result differs from that of an IntelliCage study wherein female mice around one year old performed better than males in place learning with reversal [44]. The results of the single corner restriction program for cohort B suggest that although individual variance exists among the rats, it is small enough that animals can be approximated as identical subjects for these IntelliCage experiments. Generating learning curves with aggregate cohort data is one way to reduce the impact of this variance on interpretation of cohort performance. Using AUC as a metric for comparing learning between groups is a natural extension of using linear fits on learning curves to estimate learning rate and takes full advantage of the data volume the IntelliCage offers. Task acquisition can be characterized by performance parameters—in this case, AUC—that are greater or less than the value that would be expected through chance alone, depending on the visit category. Chance C-AUC/O-AUC would be equal to the area of a right triangle with base of length 180 (number of minutes in a drinking session) and height of 0.25 (probability of visiting a correct/opposite corner at random), or 22.5. Significant session-wise differences in the appropriate direction can reflect task acquisition too, as seen with increases in C-AUC accompanied by decreases in O-AUC. These characteristics were observed for all the spatial learning programs except serial reversal, suggesting that the program is too complex for the rats to learn by 4–5 months of age. Acquisition of behavioral sequencing and serial reversal tasks in the form of diagonal shuttling and switching, respectively, has been well established as an IntelliCage paradigm for mice by Endo et al. [34]; however, the diagonal switches were originally programmed to occur automatically, independently of nosepokes, on a timescale of 4–7 3-hour drinking sessions *per switch*, making our version of the serial reversal task considerably more difficult. In general, a task that challenges the animals

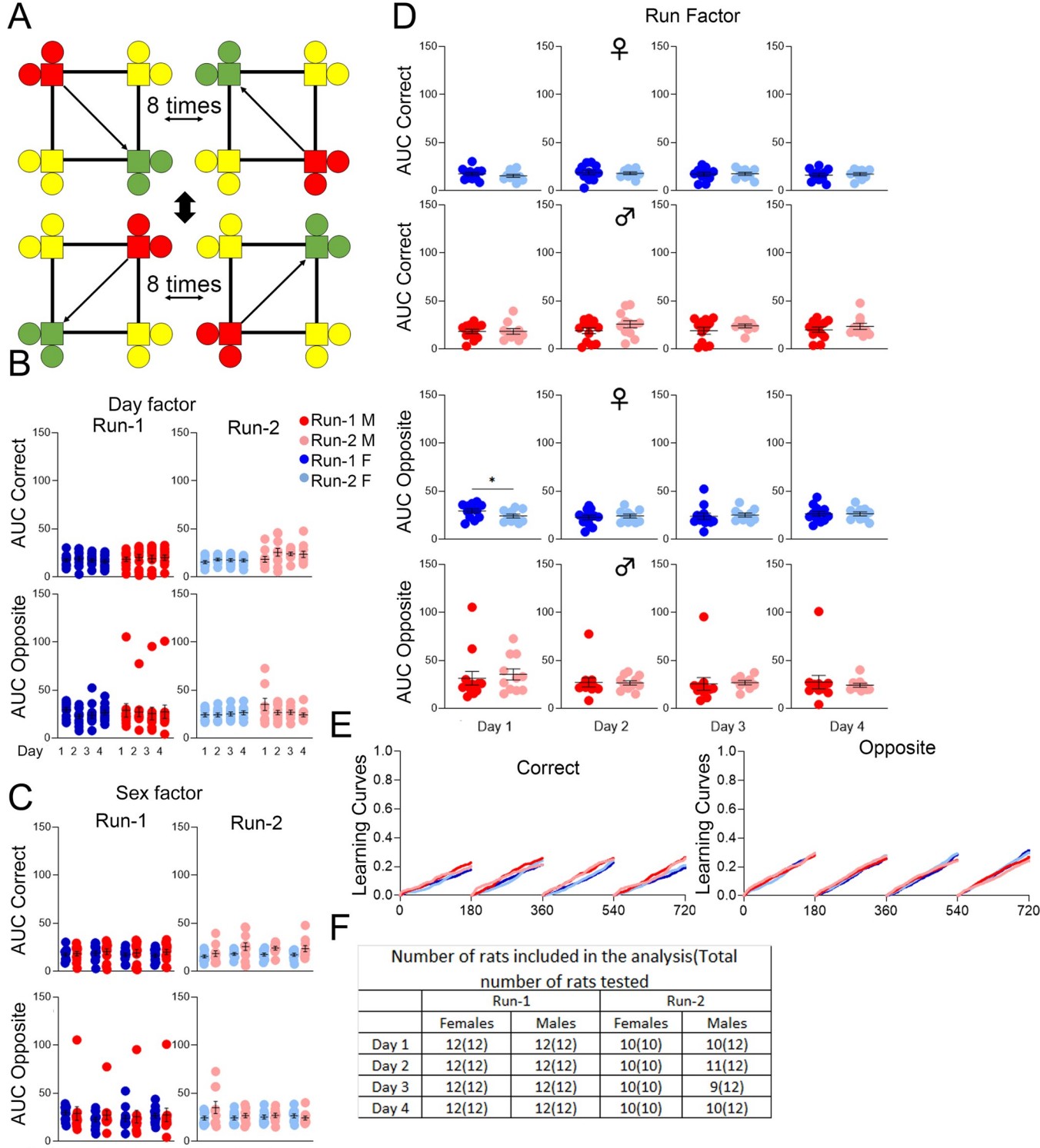

**Fig 9. Serial reversal, cohort A. (A)** Schematic of correct (green), lateral (yellow), and opposite (red) corner layouts and pattern. The starting layout depends on the initial corner assignment. **(B)** Day factor, **(C)** Sex factor and **(D)** Run factor comparisons of area under the curve (AUC) for activity curves of individual animals by sex, program day, and Run, separated by visit categories (Correct or Opposite). **(E)** Activity curves showing the fractional accumulation (y-axis) of correct visits or opposite visits over drinking session time (x-axis), reset every 180 minutes, by sex and Run. **(F)** For Run-1, 12 male and 12 female rats were tested. All 24 rats were included in the analysis of Run-1. For Run-2, 12 male and 10 female rats were tested, since two females died during the time between Runs. Two males were excluded from the analysis of days 1 and 4, one male was excluded from the analysis of day 2, and three males were excluded from the

analysis of day 3 during Run-2. These animals were excluded because they visited corners less than 25 times during those drinking sessions. All data represented as mean ± SEM (*$p < 0.05$, **$p < 0.01$, ***$p < 0.001$, ****$p < 0.0001$). See Tables 14 and 15 for statistical analysis. Males (M or ♂) are colored as red (Run-1) and light red (Run-2). Females (F or ♀) are colored as blue (Run-1) or light blue (Run-2).

**Table 14. Statistical analysis of data shown in Fig 9B and 9C for serial reversal, cohort A, Run-1, and Run-2.**

| Fig 9B and 9C | Two-way RM ANOVA | | |
|---|---|---|---|
| Correct Run-1 | **Source of Variation** | **F (DFn, DFd)** | **P** |
| | Interaction | F (3, 66) = 0.3061 | 0.8209 |
| | Day factor(9B) | F (3, 66) = 0.4277 | 0.7338 |
| | Sex factor(9C) | F (1, 22) = 0.5099 | 0.4827 |
| Correct Run-2 | **Source of Variation** | **F (DFn, DFd)** | **P** |
| | Interaction | F (3, 53) = 0.6830 | 0.5664 |
| | Day factor(9B) | F (3, 53) = 2.568 | 0.0641 |
| | Sex factor(9C) | F (1, 19) = 5.754 | 0.0269 |
| Opposite Run-1 | **Source of Variation** | **F (DFn, DFd)** | **P** |
| | Interaction | F (3, 66) = 0.4999 | 0.6837 |
| | Day factor(9B) | F (3, 66) = 2.051 | 0.1152 |
| | Sex factor(9C) | F (1, 22) = 0.06284 | 0.8044 |
| Opposite Run-2 | **Source of Variation** | **F (DFn, DFd)** | **P** |
| | Interaction | F (3, 53) = 2.117 | 0.1090 |
| | Day factor(9B) | F (3, 53) = 1.178 | 0.3271 |
| | Sex factor(9C) | F (1, 19) = 1.600 | 0.2212 |

A *p*-value less than 0.05 is considered significant.

**Table 15. Statistical analysis of data shown in Fig 9D for serial reversal, cohort A, Run comparison.**

| Fig 9D | Paired *t* tests | | | |
|---|---|---|---|---|
| Run Factor Correct | **Comparison** | **t, df** | **Summary** | **P** |
| | Female, day 1 | t = 0.1544, df = 9 | ns | 0.8807 |
| | Male, day 1 | t = 0.0315, df = 9 | ns | 0.9756 |
| | Female, day 2 | t = 0.1535, df = 9 | ns | 0.8814 |
| | Male, day 2 | t = 1.396, df = 11 | ns | 0.1902 |
| | Female, day 3 | t = 0.6986, df = 9 | ns | 0.5024 |
| | Male, day 3 | t = 0.8514, df = 8 | ns | 0.4193 |
| | Female, day 4 | t = 0.2108, df = 9 | ns | 0.8377 |
| | Male, day 4 | t = 0.9781, df = 9 | ns | 0.3536 |
| Run Factor Opposite | **Comparison** | **t, df** | **Summary** | **P** |
| | Female, day 1 | t = 2.629, df = 9 | * | 0.0274 |
| | Male, day 1 | t = 0.2406, df = 10 | ns | 0.8147 |
| | Female, day 2 | t = 0.8891, df = 9 | ns | 0.3971 |
| | Male, day 2 | t = 0.1268, df = 10 | ns | 0.9016 |
| | Female, day 3 | t = 0.6236, df = 9 | ns | 0.5483 |
| | Male, day 3 | t = 0.1706, df = 8 | ns | 0.8688 |
| | Female, day 4 | t = 0.7629, df = 9 | ns | 0.4651 |
| | Male, day 4 | t = 0.4479, df = 9 | ns | 0.6648 |

A *p*-value less than 0.05 is considered significant (*$p < 0.05$). ns = not significant.

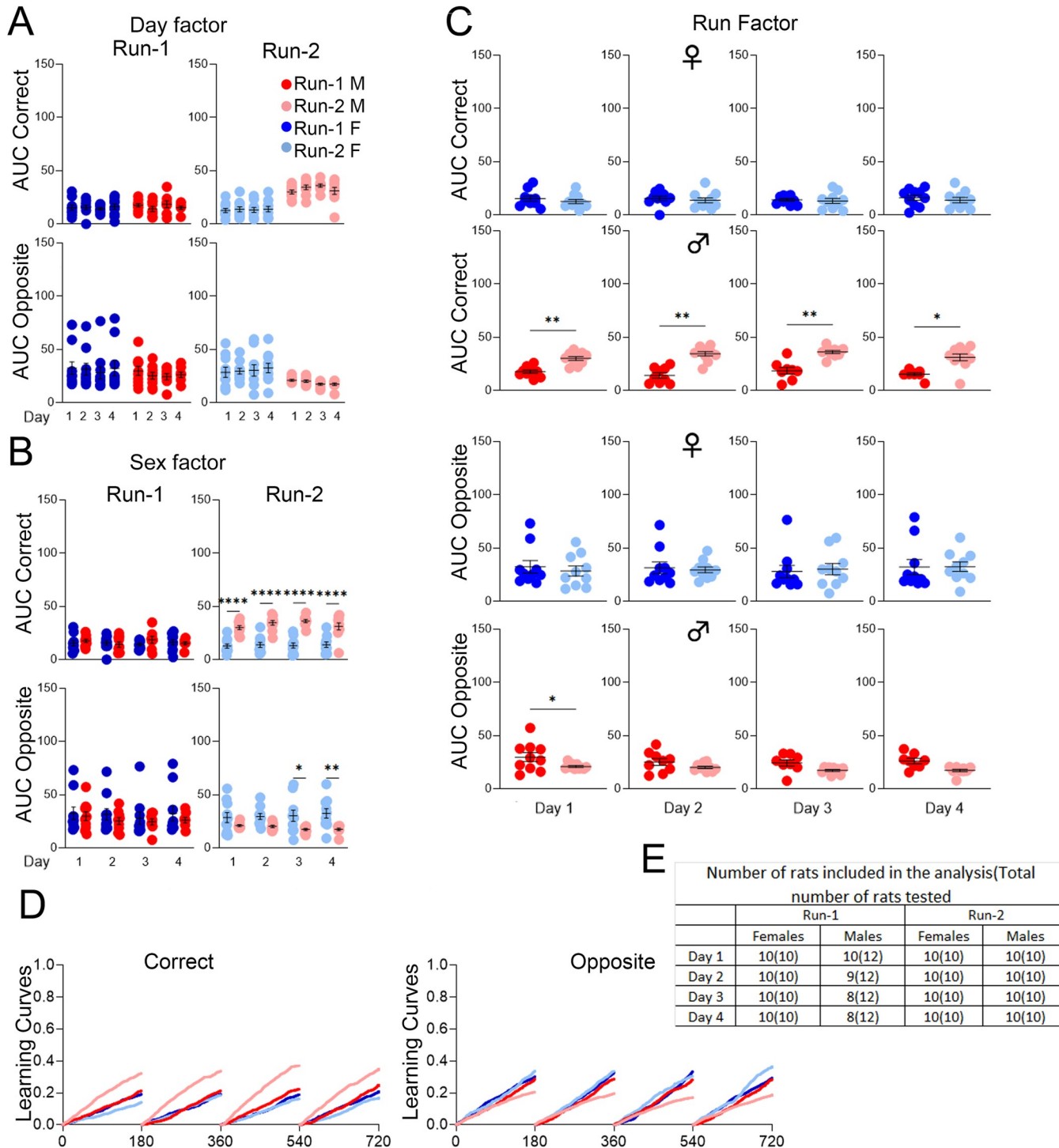

**Fig 10. Serial reversal, cohort B. (A)** Day factor, **(B)** Sex factor and **(C)** Run factor comparisons of area under the curve (AUC) for activity curves of individual animals by sex, program day, and Run, separated by visit categories (Correct, SR-C or Opposite, SR-O). **(D)** Activity curves showing the fractional accumulation (y-axis) of correct visits or opposite visits over drinking session time (x-axis), reset every 180 minutes, by sex and Run. **(E)** For Run-1, 12 male and 10 female rats were tested since 2 females died just before the timeline started. Two males were excluded from the analysis of day 1, three males were excluded from the analysis of day 2, and four males were excluded from the analysis of days 3 and 4 during Run-1. These animals were excluded because they visited corners less than 25 times during the drinking session. For Run-2, 10 male and 10 female rats were tested, since one male died during the time between Runs and one male died after place learning with corner switch. All 20 rats were included in the analysis of Run-2. All data represented as mean ± SEM ($^{*}p < 0.05$, $^{**}p < 0.01$, $^{***}p < 0.001$, $^{****}p < 0.0001$). See Tables 16 and 17 for statistical analysis. Males (M or ♂) are colored as red (Run-1) and light red (Run-2). Females (F or ♀) are colored as blue (Run-1) or light blue (Run-2).

**Table 16. Statistical analysis of data shown in Fig 10A and 10B for serial reversal, cohort B, Run-1, and Run-2.**

| Fig 10A and 10B | Mixed-effects analysis | | |
|---|---|---|---|
| Correct Run-1 | **Source of Variation** | **F (DFn, DFd)** | **P** |
| | Interaction | F (3, 49) = 1.404 | 0.2527 |
| | Day factor(10A) | F (3, 49) = 0.4026 | 0.7518 |
| | Sex factor(10B) | F (1, 18) = 0.2942 | 0.5942 |
| Correct Run-2 | **Source of Variation** | **F (DFn, DFd)** | **P** |
| | Interaction | F (3, 54) = 1.273 | 0.2928 |
| | Day factor(10A) | F (3, 54) = 1.481 | 0.2300 |
| | Sex factor(10B) | F (1, 18) = 61.77 | <0.0001 |
| | **post-hoc Sidak's multiple comp. test** | **Summary** | **Adjusted P** |
| | Female vs. Male, day 1 | **** | <0.0001 |
| | Female vs. Male, day 2 | **** | <0.0001 |
| | Female vs. Male, day 3 | **** | <0.0001 |
| | Female vs. Male, day 4 | **** | <0.0001 |
| Opposite Run-1 | **Source of Variation** | **F (DFn, DFd)** | **P** |
| | Interaction | F (3, 49) = 0.1940 | 0.9000 |
| | Day factor(10A) | F (3, 49) = 0.7972 | 0.5014 |
| | Sex factor(10B) | F (1, 18) = 0.7083 | 0.4111 |
| Opposite Run-2 | **Source of Variation** | **F (DFn, DFd)** | **P** |
| | Interaction | F (3, 54) = 0.6292 | 0.5993 |
| | Day factor(10A) | F (3, 54) = 0.05791 | 0.9815 |
| | Sex factor(10B) | F (1, 18) = 18.17 | 0.0005 |
| | **post-hoc Sidak's multiple comp. test** | **Summary** | **Adjusted P** |
| | Female vs. Male, day 3 | * | 0.0242 |
| | Female vs. Male, day 4 | ** | 0.0056 |

($^*p < 0.05$, $^{**}p < 0.01$, $^{****}p < 0.0001$). A $p$-value less than 0.05 is considered significant.

**Table 17. Statistical analysis of data shown in Fig 10C for serial reversal, cohort B, Run comparison.**

| Fig 10C | Paired $t$ tests | | | |
|---|---|---|---|---|
| Run Factor Correct | **Comparison** | **t, df** | **Summary** | **P** |
| | Female, day 1 | t = 0.8066, df = 9 | ns | 0.4407 |
| | Male, day 1 | t = 4.498, df = 8 | ** | 0.0020 |
| | Female, day 2 | t = 0.5204, df = 9 | ns | 0.6153 |
| | Male, day 2 | t = 5.283, df = 7 | ** | 0.0011 |
| | Female, day 3 | t = 0.4795, df = 9 | ns | 0.6430 |
| | Male, day 3 | t = 5.137, df = 6 | ** | 0.0021 |
| | Female, day 4 | t = 0.7748, df = 9 | ns | 0.4583 |
| | Male, day 4 | t = 3.550, df = 6 | * | 0.0121 |
| Run Factor Opposite | **Comparison** | **t, df** | **Summary** | **P** |
| | Female, day 1 | t = 0.6327, df = 9 | ns | 0.5427 |
| | Male, day 1 | t = 2.358, df = 8 | * | 0.0461 |
| | Female, day 2 | t = 0.3549, df = 9 | ns | 0.7308 |
| | Male, day 2 | t = 1.223, df = 7 | ns | 0.2608 |
| | Female, day 3 | t = 0.3089, df = 9 | ns | 0.7644 |
| | Male, day 3 | t = 1.474, df = 6 | ns | 0.1909 |
| | Female, day 4 | t = 0.0202, df = 9 | ns | 0.9843 |
| | Male, day 4 | t = 2.396, df = 6 | ns | 0.0536 |

A $p$-value less than 0.05 is considered significant ($^*p < 0.05$, $^{**}p < 0.01$). ns = not significant.

without being impossible to acquire would be ideal for identifying possible cognitive deficits in models of neurodegeneration and dementia. Task acquisition of behavioral sequencing but not serial reversal suggests that a program of intermediate difficulty using a sequence involving all four corners (in clockwise motion, for example) rather than just two in a single diagonal, might be worth testing in future studies. Yet, such a program could be affected by development of pervasive strategy (to visit each next corner) that can collapse the cognitive demand of this particular protocol.

By these measures, this study establishes a baseline spatial learning profile for $App^{h/h}$ control rats while exploring analytic methods involving aggregate cohort learning and use of AUC as a metric for task performance in the IntelliCage.

In summary, the longitudinal behavioral analysis tested here with the IntelliCage system can be useful to determine, in follow-up studies, whether knock-in rat models of FAD, LOAD, and ADRD develop aging-dependent learning and memory deficits, whether these deficits correlate with either early synaptic plasticity/transmission alterations or potential AD-like pathology, and the impact of sex.

## Author Contributions

**Conceptualization:** Hoa Pham, Tao Yin, Luciano D'Adamio.

**Data curation:** Hoa Pham, Luciano D'Adamio.

**Formal analysis:** Hoa Pham, Tao Yin, Luciano D'Adamio.

**Funding acquisition:** Luciano D'Adamio.

**Investigation:** Hoa Pham.

**Methodology:** Hoa Pham, Tao Yin, Luciano D'Adamio.

**Project administration:** Luciano D'Adamio.

**Resources:** Luciano D'Adamio.

**Software:** Hoa Pham.

**Supervision:** Luciano D'Adamio.

**Validation:** Hoa Pham, Luciano D'Adamio.

**Writing – original draft:** Hoa Pham, Tao Yin.

**Writing – review & editing:** Hoa Pham, Tao Yin, Luciano D'Adamio.

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
