## [Decision Letter · Decision Letter 0]

15 Mar 2022

PONE-D-22-01698Initial assessment of the spatial learning, reversal, and sequencing task capabilities of knock-in rats with humanizing mutations in the Aß-coding region of AppPLOS ONE

Dear Dr. D'Adamio,

Thank you for submitting your manuscript to PLOS ONE. After careful consideration by 2 Reviewers and an Academic Editor, all of the critiques of both Reviewers, especially Reviewer #2, must be addressed in detail in a revision to determine publication status. Please pay particular attention to the critiques regarding lack of validation/controls and confusing structure of the Figures as well as a general lack of description of the Intellicage. If you are prepared to undertake the work required, I would be pleased to reconsider my decision, but revision of the original submission without directly addressing the critiques of the Reviewers does not guarantee acceptance for publication in PLOS ONE. If the authors do not feel that the queries can be addressed, please consider submitting to another publication medium. A revised submission will be sent out for re-review. The authors are urged to have the manuscript given a hard copyedit for syntax and grammar.

We look forward to receiving your revised manuscript.

Kind regards,

Stephen D. Ginsberg, Ph.D.

Section Editor

PLOS ONE

Journal Requirements:

**Comments to the Author**

1. Is the manuscript technically sound, and do the data support the conclusions?

Reviewer #1: Yes

Reviewer #2: Yes

2. Has the statistical analysis been performed appropriately and rigorously? 

Reviewer #1: Yes

Reviewer #2: No

3. Have the authors made all data underlying the findings in their manuscript fully available?

Reviewer #1: Yes

Reviewer #2: No

4. Is the manuscript presented in an intelligible fashion and written in standard English?

Reviewer #1: Yes

Reviewer #2: No

5. Review Comments to the Author

Reviewer #1: The manuscript is well written, timely, and is important to establish the use of behavioral apparatuses (Intellicage) that remove variables that may contaminate results. Major concerns noted below include not validating pathology in the rat model, missing details in the methods section, and elaborating in the discussion on how this is related to already published work in the AD field related to rodent models and Intellicage testing.

Major concerns:

1) Methods: Key methods on the rat model used are missing. How did they verify the genetic background of the rats? PCR?

2) There is no validation of pathology in this rat model. If the point is that this model is superior than mouse models over-expressing APP, data on pathology needs to be presented. At the very least, measures of abeta soluble and insoluble fractions should be included and correlated with performance in the intellicage. It is also unclear if rats would have pathology at this age, so ages selected is not clear.

3) There is no mention of what the protocol is if an animal does not drink during the testing period. In various studies, animals have been shown not to drink in the Intellicage, it is a frequent occurrence. How many rats were excluded because of this? This should be elaborated in the methods section and is critically important for animal ethical standards.

4) The Discussion lacks detail. There is no discussion on how performance of the rats compares to 1) other behavioral tasks (ie., MWM, object recognition, etc.) or 2) to performance of mice in the IntellICage. There are many recent papers on this topic in the last 2 years.

5) Why were the ages selected and how do they relate to pathology?

Minor concerns:

1. The introduction can benefit from a paragraph on what is the intellicage, its background, and how it differs from standard cognitive tasks.

2. The abstract needs concluding sentences to summarize the key findings of this work and the impact to the field of AD rodent modeling.

Reviewer #2: This is an interesting and important study using humanized model of APP in rats (APP KI) carrying humanized mutations in the Ab region of APP. The main focus of the study is on utilization of IntelliCage to delineate possible cognitive phenotypes of this APP KI model.

Below are some of my comments to improve this otherwise interesting manuscript.

Suggestions/Concerns:

- The introduction seems to introduce many general topics and it might be helpful to somewhat re-structure it to align it with main focus of this particular study.

- The description of the rat model in the Methods section is not sufficient. More details are needed to guide readers' understanding which mutations are introduced and whether the levels of APP and/or Ab are affected in this model.

- Please state explicitly and visibly in the Methods section the total number of rats used for each protocol/cohort/sex as well as a number of rats per IntelliCage.

- Figures 1-2 are helpful showing the nature of the tasks used. It would be nice to put the names of different protocols in the panels and include easily accessible information on a time frame for each stage.

- The description of IntelliCage outcomes in the Methods section is convoluted and too complex to follow. Some explanations of why such outcomes were chosen with their general meaning would be helpful.

- Statistical analyses section does not address statistical analyses but rather consist of further workflow description of how to receive the outcomes.

- The major issue is the lack of the control group. From reading the Methods section, It is unclear what comparisons are of interest in this study. There is no description of what types of comparisons for which factors are planned.

- Results are presented in terms of cohorts, and it is unclear what meaning these descriptions have. For example, why the results started from Cohort B? What is the meaning of comparing one female ra to two other female rats at different passes?

- If " activity curves show.. the fractional accumulation of corner visits" as stated in Results, then the results for correct and incorrect activity curves are dependent on each other and do not need to be represented as separate outcomes.

- There are 10 multi-panel figures accompanied by more than 14 tables with stat results. How the effect of multiple comparisons in this dataset been addressed is not clear.

- The figures could be edited to present more easily perceived information. Names of the task, scheme of the task, etc.

- The discussion/intro do not place this study in any background of what has been done in this environment. The choice of the tasks is not discussed as well as the choice of the outcomes used. All of these shortcomings significantly decrease enthusiasm to this potentially interesting study.

6. PLOS authors have the option to publish the peer review history of their article (what does this mean?). If published, this will include your full peer review and any attached files.

**Do you want your identity to be public for this peer review?** For information about this choice, including consent withdrawal, please see our Privacy Policy.

Reviewer #1: No

Reviewer #2: **Yes: **Alena Savonenko

---

## [Author Response · Author response to Decision Letter 0]

30 Mar 2022

Reviewer #1: 

Major concerns:

1) Methods: Key methods on the rat model used are missing. How did they verify the genetic background of the rats? PCR?

Response: Thank you for this comment. To address these questions, we have included a new Figure (Fig. 2). In this figure we show the mutations introduced to humanize the Aβ region of rat App and the sequencing method used to verify the genotype of rats (Fig. 2A). To verify whether the protein products of the Apph/h allele contain the humanizing mutations, we performed the following experiments. 1) an ELISA assay using as detection antibody either M3.2 (a mouse monoclonal raised against the rat APP sequence between the β‐ and α‐secretase cleavage sites DAEFGHDSGFEVRHQK, which only recognize APP molecules containing the rat Aβ sequence) or 6E10 (a mouse monoclonal raised against the corresponding domain of human APP DAEFRHDSGYEVHHQK, which only recognize APP molecules containing the human Aβ sequence). This ELISA shows that Appw/w rats produce rat Aβ while Apph/h rats produce human Aβ (Fig. 2B). 2) Western blot analysis with M3.2 and 6E10. As shown in Fig. 2C, M3.2 detects APP only in Appw/w, conversely, 6E10 detects APP only in Apph/h rats. Altogether, these experiments demonstrate that products of the Apph allele, contain the humanized Aβ sequence. Methods related to the new Figure 4 have been added to the revised manuscript. We believe these experiments address the reviewer’s question. 

2) There is no validation of pathology in this rat model. If the point is that this model is superior to mouse models over-expressing APP, data on pathology needs to be presented. At the very least, measures of abeta soluble and insoluble fractions should be included and correlated with performance in the intellicage. It is also unclear if rats would have pathology at this age, so ages selected is not clear.

Response: Thank you for this question. In the original version, we did not satisfactorily explain the purpose behind the generation of this animal. We do not consider the Apph/h rats a model of late onset AD (LOAD). We are aware that recently Dr. LaFerla and Jackson Laboratory have produced a "humanized" App knock-in mouse which is being used by many as a model of LOAD. However, we consider our "humanized" App knock-in rat as a control animal to be used to determine the effect of familial genetic mutation causing early onset AD, or other genetic variant that increase the risk of LOAD. With this Apph allele, we can test the pathogenic mechanisms of familial and sporadic mutations/variants in the context of normal levels of human Aβ expression. We do not expect this model to develop AD-like pathology (especially not at these young ages). To proof or disproof this prediction, in this revised manuscript we show IHC analysis of 14 months old Apph/h rats: unsurprisingly, Apph/h rats show no signs of AD-like pathology, even at 14 moths of age (Fig. 2D). Methods related to the new Figure 2 have been added to the revised manuscript. We believe these experiments address the reviewer’s question. 

As for the ages selected, we based this selection on the evidence that LOAD, FAD and ADRD knock-in rat models that we have created show synaptic plasticity and transmission alterations already at 6-8 weeks of age. Thus, the first time point was chosen because, in future experiments we will test whether these familial and sporadic pathogenic mutations cause deficits in learning and memory as compared to control Apph/h rats starting at ~8 weeks of age, when these “AD” animals show synaptic plasticity/transmission alterations. We performed a second longitudinal test of the same animals because, in these future experiments, we intend to perform longitudinal studies in our FAD and LOAD rat model organisms. Therefore, we tested whether longitudinal studies using the IntelliCage are informative. In this revised version, we have included modifications in all sections to explain these concepts better.

3) There is no mention of what the protocol is if an animal does not drink during the testing period. In various studies, animals have been shown not to drink in the Intellicage, it is a frequent occurrence. How many rats were excluded because of this? This should be elaborated in the methods section and is critically important for animal ethical standards.

Response: Thank you for this question. We apologize for not having provided this important information in the first version of the paper. We have added a new section in the Methods entitled Inclusion/exclusion criteria to address this criticism. In addition, Details of data point exclusion by drinking session can be seen in the tables accompanying data figures.

4) The Discussion lacks detail. There is no discussion on how performance of the rats compares to 1) other behavioral tasks (i.e., MWM, object recognition, etc.) or 2) to performance of mice in the IntellICage. There are many recent papers on this topic in the last 2 years. 

Response: Thank you for this question. We have revised the discussion to address these points.

5) Why were the ages selected and how do they relate to pathology?

Response: See response to Criticism #2.

Minor concerns:

1. The introduction can benefit from a paragraph on what is the intellicage, its background, and how it differs from standard cognitive tasks.

Response: Thank you for this suggestion. This paragraph has been added to the

Introduction.” Behavioral tests consent to determine whether model organism of AD and ADRD develop learning & memory deficits. Most studies use traditional paradigms, including novel object recognition, Fear conditioning, Morris water maze, Radial arm water maze. These approaches are well established and informative. The IntelliCage system (NewBehavior AG) provides an additional method of assessing behavior in rodents. It has been used to study behavior in mouse models of human disease, including neurodegenerative and neuropsychiatric conditions such as Huntington’s disease and, notably, AD, with spatial learning and memory being among the most studied parameters. It consists of a central square home cage connected to four operant learning chambers, or corners. Every corner has two sides, each with a drinking bottle gated by a rotating door with a nosepoke sensor (Figure 1). The sides also include LEDs and air puff delivery valves as additional conditioning components. Behavioral programs are defined by the user within a visual coding platform. Subcutaneously injected transponders allow the IntelliCage to track the activity of individual animals with unique radio frequency identification tags. Among the parameters tabulated for subsequent analysis are corner visits, visit lengths, visit times, number of nosepokes per visit, and number of bottle licks per visit. This system offers a variety of advantages over standard cognitive tasks: high-throughput, unbiased data collection; minimal risk of human error; minimal perturbation of testing conditions; and uniform testing of multiple animals simultaneously in a social setting. The final point is important in the context of cognitive phenotyping because the social housing component, a distinguishing feature of this system, eliminates isolation as a confounding psychological stressor from the animals’ environment. Its stable, passive manner of data collection also mitigates stress from handling and the traumatic experience inherent in tests such as the Morris water maze.”

2. The abstract needs concluding sentences to summarize the key findings of this work and the impact to the field of AD rodent modeling.

Response: Thank you for this suggestion. The concluding sentence has been added to the Abstract. 

 

Reviewer #2: 

Suggestions/Concerns:

- The introduction seems to introduce many general topics and it might be helpful to somewhat re-structure it to align it with main focus of this particular study.

Response: Thank you for the suggestion. Based on this suggestion, we have extensively changed the Introduction. 

- The description of the rat model in the Methods section is not sufficient. More details are needed to guide readers' understanding which mutations are introduced and whether the levels of APP and/or Aβ are affected in this model.

Response: See response to Criticism #2 of Reviewer #1. A direct comparison between Aβ levels cannot be done since the detection antibodies are different (6E10 for human Aβ and M3.2 for rat Aβ). We do not use 4G8 because 4G8 can also detect P3 (the α-γ secretase mini- Aβ peptide) and cannot accurately measure Aβ. We have previously shown that APP expression levels are unchanged by the humanizing mutations.

- Please state explicitly and visibly in the Methods section the total number of rats used for each protocol/cohort/sex as well as a number of rats per IntelliCage.

Response: Thank you for this suggestion. We have added the requested information in the methods section. In addition, for each experiment we indicate the # of animals tested as well as the number of animals included in the analyses in the Figures and Figures legends. See also response to Criticism #3 of Reviewer #1.

- The description of IntelliCage outcomes in the Methods section is convoluted and too complex to follow. Some explanations of why such outcomes were chosen with their general meaning would be helpful.

Response: Thank you for this suggestion. We have added paragraphs describing the general rationale for outcomes (i.e., corner rank, AUC, activity curves, Statistical analysis of AUC) in the Methods section. 

Corner rank: "To understand better the effect of social interaction on the behavior of animals in the IntelliCage, we ranked animals via a point system based on whether an animal visits the correct corner more than other animals do during single corner restriction. This may occur with the exclusion of other animals from those corners and be considered a proxy for social dominance. As the single corner restriction program changes the correct corner every ninety minutes over two drinking sessions, we might expect a dominant animal to score highly for all corners."

Activity curves: "To visualize the activity of all the animals in each IntelliCage as a unit, we charted the fractional accumulation of correct/opposite visits over the course of each drinking session. With the resulting curves, we can qualitatively compare task performance according to drinking session, sex, and pass."

Statistical analysis of AUC: "To assess task performance quantitatively, we used the area under the activity curves of individual animals in each IntelliCage. Every correct/opposite visit during a drinking session contributes to this area; this approach accounts not only for the total fraction of correct/opposite visits but also for the rate at which they accumulate. It also takes advantage of the large volume of data the IntelliCage collects from each program such that there is no need to approximate the rate of learning with curve fitting."

- Statistical analyses section does not address statistical analyses but rather consist of further workflow description of how to receive the outcomes. 

Response: Thank you. We have listed which tests we used and the comparisons we were interested in for each part of the analysis in the Methods section.

Corner rank: "[We] compared the mean scores of animals within each IntelliCage for a given pass, performing a one-way ordinary ANOVA in Prism 9 (GraphPad, San Diego, California) followed by Tukey’s multiple comparisons tests when applicable (p < 0.05 was significant)."

AUC: "We focused on three factors in our analysis: drinking session, sex, and pass. Given a pass and program, we performed a two-way repeated measures ANOVA in Prism 9 on the data from all animals in a cohort, organized by sex and drinking session, followed by Šídák's multiple comparisons tests when applicable (p < 0.05 was significant). Specifically, we wanted to see whether there were significant session-wise differences within sex or sex differences within a given drinking session. If one or more drinking session data points were excluded for a given pass and program, mixed-effects analysis was performed instead with appropriate post-hoc tests. Paired t tests were performed to compare performance between passes within a cohort for a given program, sex, and drinking session."

- The major issue is the lack of the control group. From reading the Methods section, It is unclear what comparisons are of interest in this study. There is no description of what types of comparisons for which factors are planned.

Response: Thank you for this question. In the original version, we did not satisfactorily explain the rationale for the experimental design. In this version, we more explicitly address these points ate the end of the introduction: “In this study we tested whether the IntelliCage can be used to assess learning and memory in Apph/h rats. We assessed the spatial learning, reversal, and sequencing task performance of male and females Apph/h rats at 6-8 weeks of age (peri-adolescent rats), and again at 4-5 months of age (young adult rats). We decided to start testing peri-adolescent rats based on the evidence that our FAD, ADRD and LOAD rat models show synaptic plasticity and transmission alterations already at 6-8 weeks of age. Thus, knowing the performance of our control group at this early age is important. We performed a second test on the same animals at 4-5 months of age, to understand how the control Apph/h rats perform in longitudinal tests as young adults. We tested both male and female rats to determine whether there are any sex-dependent differences in performance. This is important because incidence rates of LOAD are greater in women than men after age 85 . In summary, this study’s goal is to establish methods using the IntelliCage system to determine, in following studies, whether our FAD, ADRD and LOAD models develop learning and memory deficits, the age of onset of these deficits, whether these deficits correlate with synaptic plasticity/transmission alterations, and the impact of sex.” We have also restructured Figures and analyses to better show these comparisons of interest.

- Results are presented in terms of cohorts, and it is unclear what meaning these descriptions have. For example, why the results started from Cohort B? What is the meaning of comparing one female ra to two other female rats at different passes?

Response: Thank you for these questions. In the methods section under "IntelliCage" we have described what we mean by "cohort" and the rationale for designing two of them. " Briefly, the program timeline was divided into three parts: (1) a period during which the animals may freely explore the IntelliCage and acclimate to a daily period of restricted water access during a time window (8:00-11:00pm) called the drinking session; (2) a period consisting of place learning and reversal programs during which every animal is assigned a drinking corner during a drinking session; and (3) a period consisting of more complex sequencing programs involving a rule that governs the designation of drinking corners based on animal activity during a drinking session. Variations in the approach toward these parts prompted the design of two parallel cohorts testing the same cognitive domains, analyzed independently. Two cohorts of Apph/h rats were studied longitudinally, A and B, housed across four IntelliCages separated by sex and cohort. Twelve rats were designated for each IntelliCage such that there would be 24 rats per cohort consisting of 12 females and 12 males each. The cohorts were run on separate program timelines, once at 6-8 weeks of age and again at 4-5 months of age (the first pass and second pass through the program timeline, respectively), as outlined in Table 2 with program descriptions in included in the Figures showing the results."

The only reason the results start with reference to cohort B is because the program whose results are being described--single corner restriction--comes before place learning and reversal (cohort A) overall on the shared timeline. Emphasis on the longitudinal aspect of this study and the rationale behind corner rank comparison should address the concern of why it is relevant to compare an animal to another across passes; in this case, it was to make a point that any significant differences in corner rank that appeared during the first pass were not significant during the second pass.

- If " activity curves show. the fractional accumulation of corner visits" as stated in Results, then the results for correct and incorrect activity curves are dependent on each other and do not need to be represented as separate outcomes.

Response: Thank you for this question. We have removed activity curves for opposite (incorrect) visits from PL, PR, and CS figures. For BS and SR figures he has removed activity curves for lateral visits and kept those for correct and opposite visits.

- There are 10 multi-panel figures accompanied by more than 14 tables with stat results. How the effect of multiple comparisons in this dataset been addressed is not clear.

Response: Thank you for this question. We are not sure we understand exactly what is being asked. But if this question relates to appropriate adjustment of p values with multiple comparisons, this is addressed in the updated statistical analysis section. The tables also mention that we used post-hoc tests addressing correction for multiple comparisons.

- Figures 1-2 are helpful showing the nature of the tasks used. It would be nice to put the names of different protocols in the panels and include easily accessible information on a time frame for each stage.

- The figures could be edited to present more easily perceived information. Names of the task, scheme of the task, etc.

Responses: Thank you for these questions. We have changed the figures to include the abbreviation of the task name in the inclusion/exclusion table. We have also moved the panels from previous Figure 2 describing the program to each data figure.

- The discussion/intro do not place this study in any background of what has been done in this environment. The choice of the tasks is not discussed as well as the choice of the outcomes used. All of these shortcomings significantly decrease enthusiasm to this potentially interesting study.

Response: Thank you for these questions. The updated introduction partially addresses this, "[The IntelliCage] has been used to study behavior in mouse models of human disease, including neurodegenerative and neuropsychiatric conditions such as Huntington’s disease and, notably, AD, with spatial learning and memory being among the most studied parameters. [...] A larger version of the IntelliCage system developed for rats has been used for studying Huntington’s disease(40); the effect of GABAB receptors in the insula on recognition memory(36); and deficits in spatial learning and memory following post-weaning social isolation(41)." The choice of outcomes is discussed in the methods section.

In addition, we have changed the discussion to further address this point.

---

## [Decision Letter · Decision Letter 1]

18 Apr 2022

PONE-D-22-01698R1Initial assessment of the spatial learning, reversal, and sequencing task capabilities of knock-in rats with humanizing mutations in the Aß-coding region of AppPLOS ONE

Dear Dr. D'Adamio,

Thank you for resubmitting your work to PLOS ONE. Please make the corrections posed by Reviewer #2 so I can render a decision on this manuscript.

We look forward to receiving your revised manuscript.

Kind regards,

Stephen D. Ginsberg, Ph.D.

Section Editor

PLOS ONE

Journal Requirements:

**Comments to the Author**

1. If the authors have adequately addressed your comments raised in a previous round of review and you feel that this manuscript is now acceptable for publication, you may indicate that here to bypass the “Comments to the Author” section, enter your conflict of interest statement in the “Confidential to Editor” section, and submit your "Accept" recommendation.

Reviewer #1: All comments have been addressed

Reviewer #2: All comments have been addressed

2. Is the manuscript technically sound, and do the data support the conclusions?

Reviewer #1: Yes

Reviewer #2: Yes

3. Has the statistical analysis been performed appropriately and rigorously? 

Reviewer #1: Yes

Reviewer #2: Yes

4. Have the authors made all data underlying the findings in their manuscript fully available?

Reviewer #1: Yes

Reviewer #2: Yes

5. Is the manuscript presented in an intelligible fashion and written in standard English?

Reviewer #1: Yes

Reviewer #2: Yes

6. Review Comments to the Author

Reviewer #1: Thank you for addressing all raised comments by this reviewer. 

Reviewer #2: The revised version of the manuscript is much improved. I have some comments that I believe are easily addressable:

Corner rank: The interpretation of a corner task as a measure of social dominance, although it is very interesting data, is not supported by data on social hierarchy. It should be discussed as a limitation of the study.

Activity curves: The name of this variable (activity curves) is misleading as this variable measures the fraction of correct responses (relative to visits to the opposite corner) rather than "activity", a word that strongly associates with motor/exploratory activity. The choice of this measure (proportion of correct to opposite corner) to characterize cognitive performance is not clearly explained. Why the visits to two other corners were excluded is not explained.

The analyses of reversal learning would benefit from analyses of correct vs previously baited corners.

The idea that "Apph/h rats constitute control animals" should be proven by data showing comparison of this model to wt rats. For example, we do know that presence of wild type human tau in mice does have consequences on a number of biological levels. This should be discussed as a limitation/future direction of the study.

Discussion section is much improved. A particular discussion on chance levels in different tasks is a particularly important addition. I would suggest introducing definitions of chance level for appropriate variables (including AUC) in the Method section, and, what would be really great, putting these levels as a line in appropriate figures. The issue of chance levels is especially significant for this study as its goal is to prove that rats learned the tasks.

In the discussion, it is stated that "...a program of intermediate difficulty using a sequence involving all four corners (in clockwise motion, for example) rather than just two in a single diagonal, might be worth testing in future studies." Although the main idea here (that a task of intermediate difficulty would be the most useful) sounds great, a particular example (visiting all four corners in clockwise motion) might be easily affected by development of pervasive strategy (to visit each next corner) that can collapse the cognitive demand of this particular protocol.

Minor:

- abbreviations should be explained before their first use - a KI approach in mice...

- a statement in the Introduction that " ...The KI approach ... makes no assumption about pathogenic mechanisms (except the unbiased genetic one)" is rather unclear and not needed.

- The figures are improved; but there is still a lot of information that is unreasonably hidden. For example, if FP = 1st pass, SP = 2nd pass, the figures would be much more readable if "Run 1" and "Run 2" would be used instead of FP and SP. The same could be said about other abbreviations (in the figures and Tables) which could be easily avoided altogether (BS-O, BS-C, SR, ...). While these comments may sound like non-essential for investigators who worked with these tasks, the relative novelty of ItelliCage, tasks and variables to overwhelming majority of other researchers makes it very important to spent time and make data presentation as clear as possible.

7. PLOS authors have the option to publish the peer review history of their article (what does this mean?). If published, this will include your full peer review and any attached files.

**Do you want your identity to be public for this peer review?** For information about this choice, including consent withdrawal, please see our Privacy Policy.

Reviewer #1: No

Reviewer #2: **Yes: **Alena Savonenko

---

## [Author Response · Author response to Decision Letter 1]

19 Apr 2022

Q: Corner rank: The interpretation of a corner task as a measure of social dominance, although it is very interesting data, is not supported by data on social hierarchy. It should be discussed as a limitation of the study.

R: We have eliminated any reference to this test as a measure of social dominance. The methods now read

“Corner rank comparison. To understand better the effect of social interaction on the behavior of animals in the IntelliCage, we ranked animals via a point system based on whether an animal visits the correct corner more than other animals do during single corner restriction. This may occur with the exclusion of other animals from those corners, which may bear upon the performance of paired animals in subsequent tests.”

Q: Activity curves: The name of this variable (activity curves) is misleading as this variable measures the fraction of correct responses (relative to visits to the opposite corner) rather than "activity", a word that strongly associates with motor/exploratory activity. The choice of this measure (proportion of correct to opposite corner) to characterize cognitive performance is not clearly explained. Why the visits to two other corners were excluded is not explained.

R: We have changed the name activity curves into Learning curves.

As for measuring the correct versus opposite, we realize that the writing in the methods was confusing. The sections describing the methods and rationale now read:

“Learning curves. To visualize learning of all the animals in each IntelliCage as a unit, we charted the fractional accumulation of correct visits (also opposite visits for the Behavioral sequencing and Serial reversal tasks) over the course of each drinking session. With the resulting curves, we can qualitatively compare task performance according to drinking session, sex, and Run. We followed this workflow to produce the learning curves for each program: 1. For each subset of rats by sex, cohort, and Run (e.g., cohort A males in their Run-1), count the total number of visits those rats made for each drinking session. For a 3-day program, there should be 3 totals for a given subset. 2. For each subset as described in step 1, tabulate the fractional accumulation of correct (and opposite) visits over time for each drinking session, adding to each fraction, starting from 0, the value of 1 divided by the associated total count for that subset and drinking session each time a correct (or opposite) visit occurs, and 0 otherwise.”

“Statistical analysis of area under the curves (AUC). To assess task performance quantitatively, we used the area under the learning curves of individual animals in each IntelliCage. Every correct visit during a drinking session contributes to this area; this approach accounts not only for the total fraction of correct visits but also for the rate at which they accumulate. It also takes advantage of the large volume of data the IntelliCage collects from each program such that there is no need to approximate the rate of learning with curve fitting. For the Behavioral sequencing and the Serial reversal tasks we also calculated the AUC for the opposite corner, since the opposite corner represents the previously correct corner. Thus, calculations of these two areas indicates the speed by which a rat learns the rules regulating alternation of correct to previously correct corners.”

As for the analysis of all for corners (i.e. the incorrect corners), reviewer 2 pointed out during the first round of revision, the following:

If " activity curves show.. the fractional accumulation of corner visits" as stated in Results, then the results for correct and incorrect activity curves are dependent on each other and do not need to be represented as separate outcomes.

Based on this comment we eliminated the analysis of the other corners when revising the manuscript.

Q: The analyses of reversal learning would benefit from analyses of correct vs previously baited corners.

R: Thank you for the suggestion but this analysis, which is conceptually identical to the analysis of Opposite in Behavioral sequencing and Serial reversal, is not informative in this case.

Q: The idea that "Apph/h rats constitute control animals" should be proven by data showing comparison of this model to wt rats. For example, we do know that presence of wild type human tau in mice does have consequences on a number of biological levels. This should be discussed as a limitation/future direction of the study.

R: We have added this sentence to the introduction.

“Whether expression of human Aβ is, per se’, sufficient to impact behavior will be addressed in future studies comparing Apph/h to Appw/w rats.”

Q: Discussion section is much improved. A particular discussion on chance levels in different tasks is a particularly important addition. I would suggest introducing definitions of chance level for appropriate variables (including AUC) in the Method section, and, what would be really great, putting these levels as a line in appropriate figures. The issue of chance levels is especially significant for this study as its goal is to prove that rats learned the tasks.

R: Thank you for the suggestion but we think that the chance level is sufficiently explained in the discussion and very intuitive for a system with 4 options.

Q: In the discussion, it is stated that "...a program of intermediate difficulty using a sequence involving all four corners (in clockwise motion, for example) rather than just two in a single diagonal, might be worth testing in future studies." Although the main idea here (that a task of intermediate difficulty would be the most useful) sounds great, a particular example (visiting all four corners in clockwise motion) might be easily affected by development of pervasive strategy (to visit each next corner) that can collapse the cognitive demand of this particular protocol.

R: We have added this sentence to the discussion.

“Yet, such a program could be affected by development of pervasive strategy (to visit each next corner) that can collapse the cognitive demand of this particular protocol.”

Minor:

Q: - abbreviations should be explained before their first use - a KI approach in mice...

R: We have added the KI abbreviation in the Abstract

.

Q: - a statement in the Introduction that " ...The KI approach ... makes no assumption about pathogenic mechanisms (except the unbiased genetic one)" is rather unclear and not needed.

R: The sentence has been deleted

Q: - The figures are improved; but there is still a lot of information that is unreasonably hidden. For example, if FP = 1st pass, SP = 2nd pass, the figures would be much more readable if "Run 1" and "Run 2" would be used instead of FP and SP. The same could be said about other abbreviations (in the figures and Tables) which could be easily avoided altogether (BS-O, BS-C, SR, ...). While these comments may sound like non-essential for investigators who worked with these tasks, the relative novelty of ItelliCage, tasks and variables to overwhelming majority of other researchers makes it very important to spent time and make data presentation as clear as possible.

R: First pass and second pass have need changed to Run-1 and Run-2 in text and figure.

Abbreviations have been eliminated.

---

## [Editor Report · Decision Letter 2]

22 Apr 2022

Initial assessment of the spatial learning, reversal, and sequencing task capabilities of knock-in rats with humanizing mutations in the Aß-coding region of App

PONE-D-22-01698R2

Dear Dr. D'Adamio,

We’re pleased to inform you that your manuscript has been judged scientifically suitable for publication and will be formally accepted for publication once it meets all outstanding technical requirements.

Kind regards,

Stephen D. Ginsberg, Ph.D.

Section Editor

PLOS ONE

---

## [Editor Report · Acceptance letter]

25 Apr 2022

PONE-D-22-01698R2 

Initial assessment of the spatial learning, reversal, and sequencing task capabilities of knock-in rats with humanizing mutations in the Aβ-coding region of *App*

Dear Dr. D'Adamio:

I'm pleased to inform you that your manuscript has been deemed suitable for publication in PLOS ONE. Congratulations! Your manuscript is now with our production department. 

Kind regards, 

on behalf of

Dr. Stephen D. Ginsberg 

Section Editor

PLOS ONE